# Facet-selective growth of halide perovskite/2D semiconductor van der Waals heterostructures for improved optical gain and lasing

Liqiang Zhang[1,5], Yiliu Wang [2,5], Anshi Chu[2,5], Zhengwei Zhang [3], Miaomiao Liu[1], Xiaohua Shen[1], Bailing Li[1], Xu Li[2], Chen Yi[2], Rong Song[1], Yingying Liu[1], Xiujuan Zhuang[4] & Xidong Duan [1] ✉

The tunable properties of halide perovskite/two dimensional (2D) semiconductor mixed-dimensional van der Waals heterostructures offer high flexibility for innovating optoelectronic and photonic devices. However, the general and robust growth of high-quality monocrystalline halide perovskite/2D semiconductor heterostructures with attractive optical properties has remained challenging. Here, we demonstrate a universal van der Waals heteroepitaxy strategy to synthesize a library of facet-specific single-crystalline halide perovskite/2D semiconductor (multi)heterostructures. The obtained heterostructures can be broadly tailored by selecting the coupling layer of interest, and can include perovskites varying from all-inorganic to organic-inorganic hybrid counterparts, individual transition metal dichalcogenides or 2D heterojunctions. The $CsPbI_2Br/WSe_2$ heterostructures demonstrate ultrahigh optical gain coefficient, reduced gain threshold and prolonged gain lifetime, which are attributed to the reduced energetic disorder. Accordingly, the self-organized halide perovskite/2D semiconductor heterostructure lasers show highly reproducible single-mode lasing with largely reduced lasing threshold and improved stability. Our findings provide a high-quality and versatile material platform for probing unique optoelectronic and photonic physics and developing further electrically driven on-chip lasers, nanophotonic devices and electronic-photonic integrated systems.

The emergence of two-dimensional (2D) semiconductor and 2D van der Waals (vdW) heterostructures have garnered tremendous attention to use them as atomically thin building blocks for designing nanoelectronics, optoelectronics and nanophotonics[1–5]. However, the atomically thin nature of such building blocks has leaded to dreadfully parasitic drawbacks: insufficient absorption of above-bandgap incident photons to carrier generation and incapacity of sustaining strong light-matter interaction for generating optical gain. These hurdles are

[1]Hunan Provincial Key Laboratory of Two-Dimensional Materials, State Key Laboratory for Chemo/Biosensing and Chemometrics, College of Chemistry and Chemical Engineering, Hunan University, Changsha, P. R. China. [2]Key Laboratory for Micro-Nano Optoelectronic Devices of Ministry of Education, School of Physics and Electronics, Hunan University, Changsha, China. [3]Hunan Key Laboratory of Nanophotonics and Devices, School of Physics, Central South University, Changsha, Hunan, P. R. China. [4]College of Semiconductors (College of Integrated Circuits), Hunan University, Changsha, Hunan, P. R. China. [5]These authors contributed equally: Liqiang Zhang, Yiliu Wang, Anshi Chu. ✉e-mail: xidongduan@hnu.edu.cn

critically detrimental to widespread adoption in optoelectronics and photonics, such as photovoltaics and nanolasers, respectively. To address these intrinsic shortcomings, vdW integration inspired by the self-passivated, dangling-bond-free surface of 2D materials offers an alternative way of integrating highly disparate materials with different dimensionality to form mixed-dimensional vdW heterostructures. They can synergistically tailor unique optoelectronic and photonic properties for mitigating the intrinsic shortcomings of the individual component, thus innovating ubiquitous optoelectronics with enhanced performances and functionalities[6–10]. Among them, halide perovskite/2D semiconductor mixed-dimensional vdW heterostructures stand out, because halide perovskites possess a unique combined optoelectronic and photonic properties, such as large absorption coefficient and refractive index, low trap density and Urbach energy, high photoluminescence quantum yield and bandwidths, balanced ambipolar charge-carrier transport, unique photon cycling, tunable bandgap and slow Auger recombination rate. These properties collectively provide the effective remedy for both 2D optoelectronic and photonic devices[11–18]. Although exciting opportunity and promise, the realization of high-quality monocrystalline halide perovskite/2D semiconductor mixed-dimensional heterostructures with exceptional optical gain properties towards photonic lasing application is still at a premature stage.

Despite recent efforts in the assembly of halide perovskite/2D semiconductor mixed-dimensional vdW heterostructures, the currently reported heterostructures have focused largely on wide-bandgap halide perovskites-based heterostructures for optoelectronic applications. They were achieved through an arduous micromechanical exfoliation-stacking process, vapor-phase-assisted intercalated transformation, and solution-phase assembly[19–27]. These routes usually suffer from limited yield and reproducibility, transfer-process-related obstacles (solvent exposure), inferior crystal quality, defective polycrystalline structures, and incompatibility with narrow-bandgap perovskite integration. The overall difficulties come from two aspects, one is the ultrasensitivity of structural distortions and phase transition under external stresses due to the ionic-bound soft-lattice crystal framework. Another problem lies in the complex reactivity of iodine, which typically results in the generation of surface defects and energetic disorder[28–30]. Furthermore, due to the huge difference in chemical composition and bonding polarity, the mismatched reactivity tolerance between multi-element ionic halide perovskite and 2D semiconductors further exacerbates the difficulty in monolithic integration (Supplementary Fig. 1). Thus far, the general and robust growth of single-crystalline narrow-bandgap halide perovskite/2D semiconductor heterostructures with minimized energetic disorder for high optical gain and photonic lasing application has remained difficult to achieve.

Herein we synthesized a library of high-quality single-crystalline halide perovskite/2D semiconductor heterostructures by a straightforward van der Waals heteroepitaxial approach. The halide perovskite/2D semiconductor heterostructures can be tailored on-demand by coupling specific perovskite epilayer and 2D semiconductor. Halide perovskites can vary from all-inorganic to organic-inorganic hybrid counterparts, and 2D semiconductors can range from different single 2D semiconductors to 2D van der Waals heterostructure, thereby enabling programmable heterostructures with well-defined spatial arrangements of different components and tunable properties for synergistically optimizing material properties and device performances. The epitaxial perovskites show high facet and alignment selectivity, probably originating from thermodynamically favorable interfacial formation energies and their degenerate state formation at the three-fold symmetry of the underlying monolayer semiconductor. Furthermore, the van der Waals epitaxy method has general applicability to both CMOS-compatible substrates (SiO$_2$/Si) and photonic-compatible platforms (Si and LiNbO$_3$). The weak van der Waals interaction produces incommensurate/incoherent in-plane lattices at the heterointerface, thereby enabling an alternative bond-free integration with minimized mismatch-induced strain and defects. As demonstrated by experimental results, the van der Waals epitaxial perovskite semiconductors show a substantially reduced defect density and homogeneous energy landscape, which yield enhanced optical gain properties and ultralow-threshold and stable single-mode lasers. Our findings expand the family of van der Waals heterostructures and highlight the advantages and prospects of halide perovskite/2D semiconductor mixed-dimensional heterostructures for on-chip light sources and integrated optoelectronics devices.

## Results

### Halide perovskite/2D semiconductor heterostructures

The epitaxial growth of soft-lattice halide perovskite/2D semiconductor heterostructures involves two steps: fabrication of atomically thin 2D semiconductors (WS$_2$, MoS$_2$, MoSe$_2$, and WSe$_2$ shown in Fig. 1a) on SiO$_2$/Si substrate, and direct epitaxy growth of halide perovskites using physic vapor deposition method, the detailed experimental procedures are presented in Methods. The 2D semiconductor domain features a perfect equilateral triangle morphology, and the epitaxial perovskites feature well-defined square/rectangle morphology and are automatically aligned with the edges of monolayer WSe$_2$, highlighted by colored dotted lines (Fig. 1b), strongly suggesting a unique and robust epitaxial relationship between upper perovskites and bottom 2D semiconductors. In striking contrast to the epitaxial growth on 2D WSe$_2$, the perovskites grown on bare SiO$_2$/Si substrate under the same growth conditions show 3D micropyramid and truncated micropyramid morphologies, high nucleation density, and random orientations (Fig. 1c). These results indicate that van der Waals surface of monolayer WSe$_2$ has the strong guiding/modulating ability to the growth of halide perovskite through altering growth kinetics and dynamics, which will be shown in the following. Taking advantage of the diversity of halide perovskites and 2D semiconductors, this direct epitaxy approach can be generalized for the successful experimental realization of a library of high-quality monocrystalline halide perovskite/2D semiconductor heterostructures (Fig. 1d), with the halide perovskites ranging from all-inorganic to hybrid organic-inorganic perovskites and the 2D semiconductor ranging from single monolayer semiconductors to 2D van der Waals heterostructure, including CsPbI$_3$ (CsPbI$_2$Br, CH$_3$NH$_3$PbI$_3$)/WSe$_2$ (WS$_2$, MoS$_2$ and MoSe$_2$) heterostructures and CsPbI$_3$/SnS$_2$/WSe$_2$ multi-heterostructures. In addition to the growth compatibility with the silicon-based CMOS-compatible substrate (SiO$_2$/Si), the van der Waals epitaxy method has general applicability to photonic-compatible platforms, such as Si and LiNbO$_3$, shown in Supplementary Fig. 2. The heterostructures are universally capable of robust and scalable growth on the ready-to-use photonic substrates, and the epitaxial perovskites still followed the crystal facet/alignment-specific growth habits. This universal substrate-compatible monolithic integration of halide perovskite/2D semiconductor heterostructures shows great promise for on-chip photonic device applications. The monolayer nature and stability of WSe$_2$ after perovskite epitaxy were systematically characterized by using atomic force microscope, Raman spectroscopy, photoluminescence (PL) and time-resolved photoluminescence (TRPL) (Supplementary Figs. 3, 4). Figure 2a displays the corresponding energy levels of conduction band and valence band of the perovskites and 2D semiconductors measured by ultraviolet photoemission spectroscopy (Supplementary Fig. 5). The energetic band alignments of resulting heterostructures can be on-demand switchable from staggered alignment (type-II) to straddling alignment (type-I), implying great potentials in designing light-harvesting optoelectronics, such as solar cells and self-powered photodetectors, as well as light-emitting devices such as LEDs and solid-state lasers.

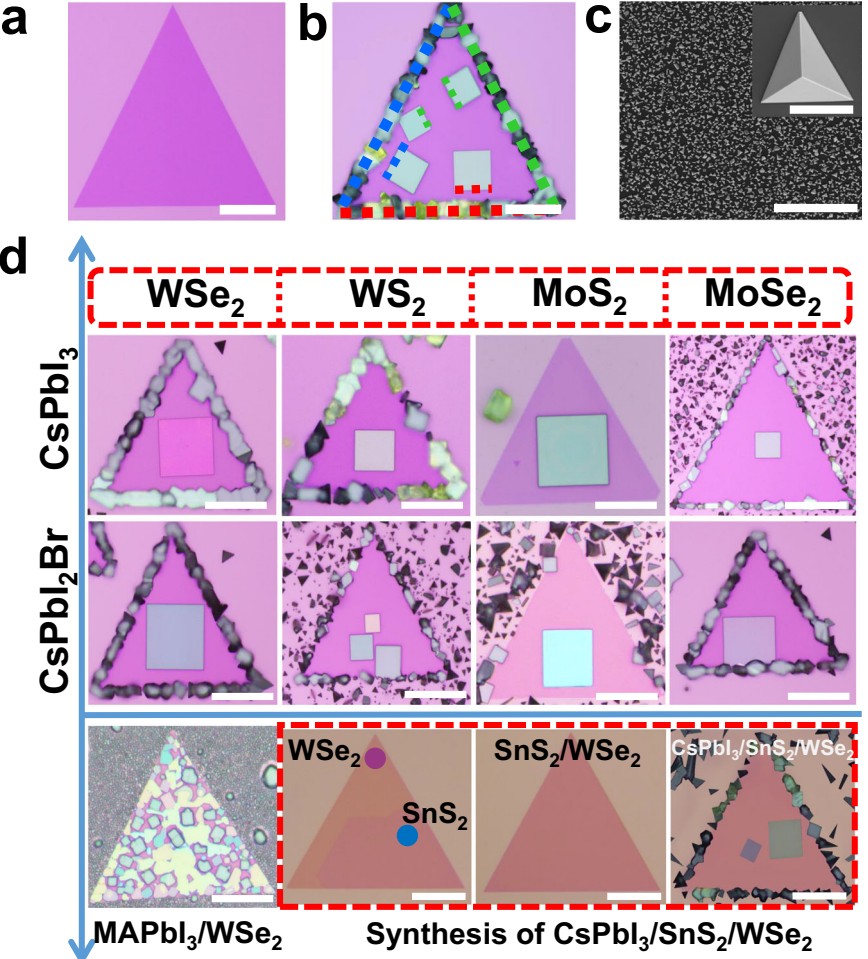

**Fig. 1 | Epitaxial growth of halide perovskite/2D semiconductor hetero-structures.** **a** Optical image of monolayer WSe$_2$ fabricated on SiO$_2$/Si substrate. **b** Optical image of CsPbI$_3$/WSe$_2$ heterostructure, the red/green/blue dotted lines on the edges of monolayer WSe$_2$ and epitaxial perovskites highlight the corresponding orientation relationship between them. Scale bars, 50 μm (**a**) and 30 μm (**b**). **c** Scanning electron microscope (SEM) images of archetypal perovskite crystals directly grown on SiO$_2$/Si substrate, scale bar, 50 μm. The inset shows the typical pyramid morphology, scale bar, 10 μm. **d** A library of halide perovskite/2D semiconductor (multi)heterostructures, scale bar, 30 μm.

Energy dispersive spectroscopy (EDS) analysis and X-ray photoelectron spectroscopy (XPS) characterization further demonstrated high phase purity and chemical compositional uniformity of epitaxial CsPbI$_3$ film (Supplementary Figs. 6, 7). The PL spectra of the representative heterostructures, isolated perovskites with equal or comparable thickness of epitaxial crystal were shown in Fig. 2b–d. The PL characteristic peaks of CsPbI$_3$ in CsPbI$_3$/WSe$_2$ and CsPbI$_3$/SnS$_2$/WSe$_2$ center at about 702 nm and exhibit remarkable quenching compared with that of the isolated one, which originates from non-radiative decay of charge carriers to the ground state caused by the efficient charge transfer at heterointerface, suggesting strong interfacial coupling. Nevertheless, in the case of CsPbI$_2$Br/WSe$_2$ heterostructure, the PL signal of CsPbI$_2$Br shows analogous quenching effect but that of monolayer WSe$_2$ was slightly enhanced, which evidenced that the singlet energy transfer from halide perovskite to the exciton continuum higher above the excitonic states in monolayer WSe$_2$ takes place. The enhanced PL signal of WSe$_2$ peaked at 782 nm, showing a little red-shift compared to bare WSe$_2$ at 776 nm, which most likely stems from bandgap renormalization owing to the high dielectric environment created by the upper perovskite layer[31]. The steady-state PL responses are well consistent with the bandgap alignment characters, reflecting flexible tunability to tailor interfacial energetics for functionality-oriented material design.

## Epitaxial mechanism

To probe the mechanism of oriented epitaxial growth of halide perovskites on 2D semiconductors, we selected the CsPbI$_3$/WSe$_2$ heterostructure as a model system for simplicity. The lattice structure of the monolayer WSe$_2$ triangular domain was characterized by high-angle annular dark-field scanning-transmission electron microscopy (HAADF-STEM) and selected area electron diffraction (SEAD), as shown in Fig. 3a, b, from which the lattice spacing of (1000) plane is determined to be 2.81 Å. The hexagonal lattice is visible and divided into the W (brightest spots) and Se (dimmer spots) sublattices (inset in Fig. 3a), thus reducing the hexagonal lattice from six-fold to three-fold symmetry. Correspondingly, the ordered six [$\bar{1}$100] diffraction spots are split into two families: ka = {($\bar{1}$100), (10$\bar{1}$0), (0$\bar{1}$10)} and k$_b$ = -ka, which indicates that monolayer WSe$_2$ has a hexagonal lattice structure with three-fold symmetry, similar to other monolayer transition metal dichalcogenides such as MoS$_2$ and WS$_2$[32–34]. Figure 3c schematically illustrates the structural motif of monolayer WSe$_2$, which belongs to the non-centrosymmetric and non-polar $D_{3h}$ point group. The yellow planes and the black pole in the monolayer lattice reflect the mirror planes and the threefold rotation axis (marked by red lines), respectively. The crystal phase of epitaxial CsPbI$_3$ perovskites was identified by X-ray diffraction (XRD, Fig. 3d). A careful analysis of the peak positions, relative intensity ratios, and the splitting of the XRD patterns with the help of Powder Diffraction File cards

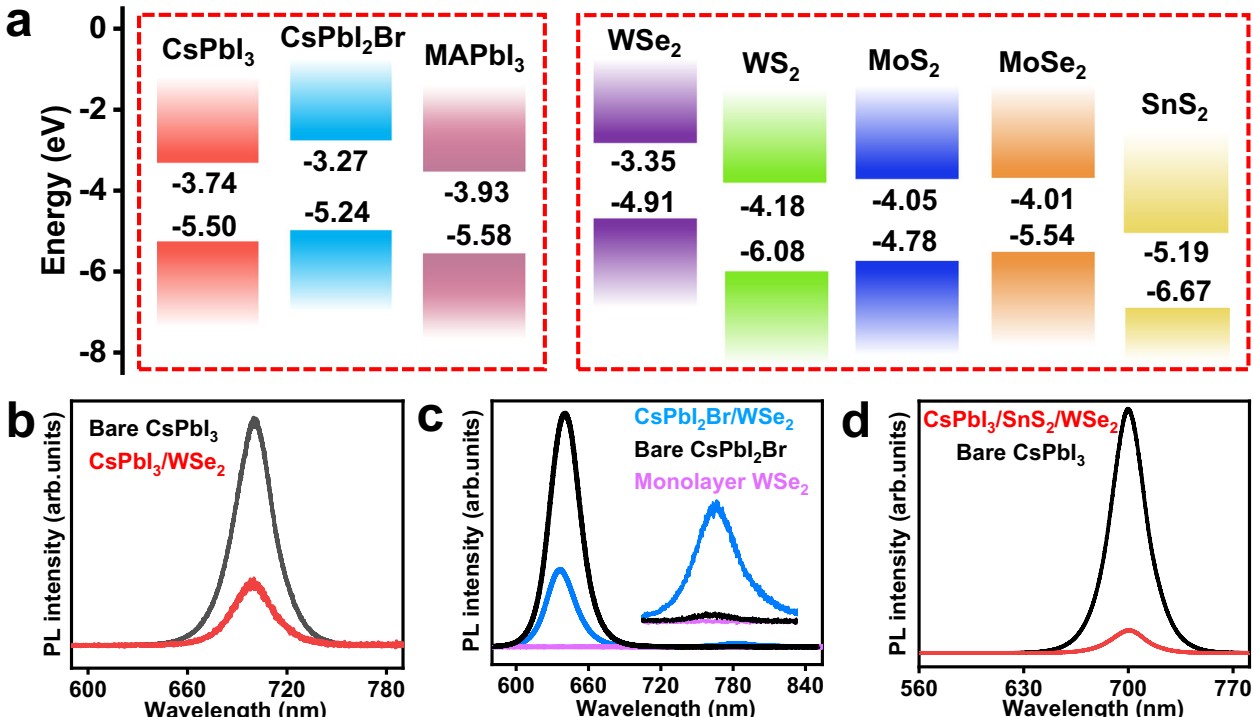

**Fig. 2 | Interfacial energetics of epitaxial heterostructures. a** Corresponding energy levels of the conduction band and valence band of the halide perovskites and 2D semiconductors used to synthesize the heterostructures. **b** Representative photoluminescence (PL) spectra of $CsPbI_3/WSe_2$ heterostructure. **c** Representative PL spectra of $CsPbI_2Br/WSe_2$ heterostructure, inset shows the corresponding magnified spectra between 750 nm to 820 nm. **d** Representative PL spectra of $CsPbI_3/SnS_2/WSe_2$ multi-heterostructure.

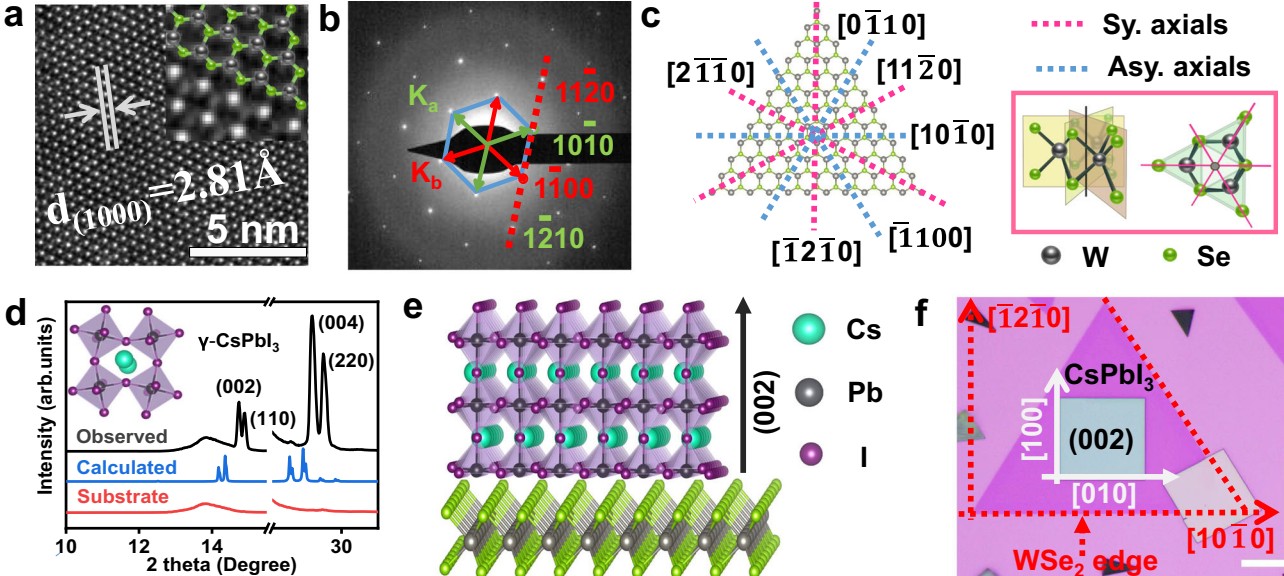

**Fig. 3 | Atomic structural motifs of monolayer $WSe_2$ and epitaxial $CsPbI_3$.**
**a** High-angle annular dark-field scanning-transmission electron microscopy ((HAADF-STEM)) of monolayer $WSe_2$. The inset is a magnified HAADF-STEM image. The bright spots are tungsten atoms, the grey spots are two stacked selenium atoms. The lattice is composed of hexagonal rings alternating tungsten (black spheres) and selenium (green spheres) sites, which are overlaid from the top view. **b** The selected area electron diffraction (SEAD) pattern of monolayer $WSe_2$. The red and green arrows represent two different families of spots, $k_b = -ka$. The blue solid lines highlight the SEAD pattern, and the red dashed line corresponds to the reciprocal lattices. **c** Threefold rotational symmetry in the atomic structure of

monolayer $WSe_2$ and its schematic illustration. The red/blue dashed lines indicate the threefold rotation axis and the black pole at the cross-section of the planes shown in the red box displays the corresponding mirror planes. Sy. Axials: Symmetrical axials; Asy. Axials: Asymmetrical axials. **d** X-ray diffraction (XRD) pattern of epitaxial $CsPbI_3$. Inset is the corresponding atomic structure of $CsPbI_3$. **e** Atomic structure of epitaxial heterostructures. **f** Possible epitaxial landscape for growth of $CsPbI_3$ on monolayer $WSe_2$. The red dashed lines are used to mark the edges of $WSe_2$. The red dashed arrows represent the crystallographic orientation of monolayer $WSe_2$, the white dashed lines represent the crystallographic orientation of epitaxial $CsPbI_3$. Scale bar, 10 μm.

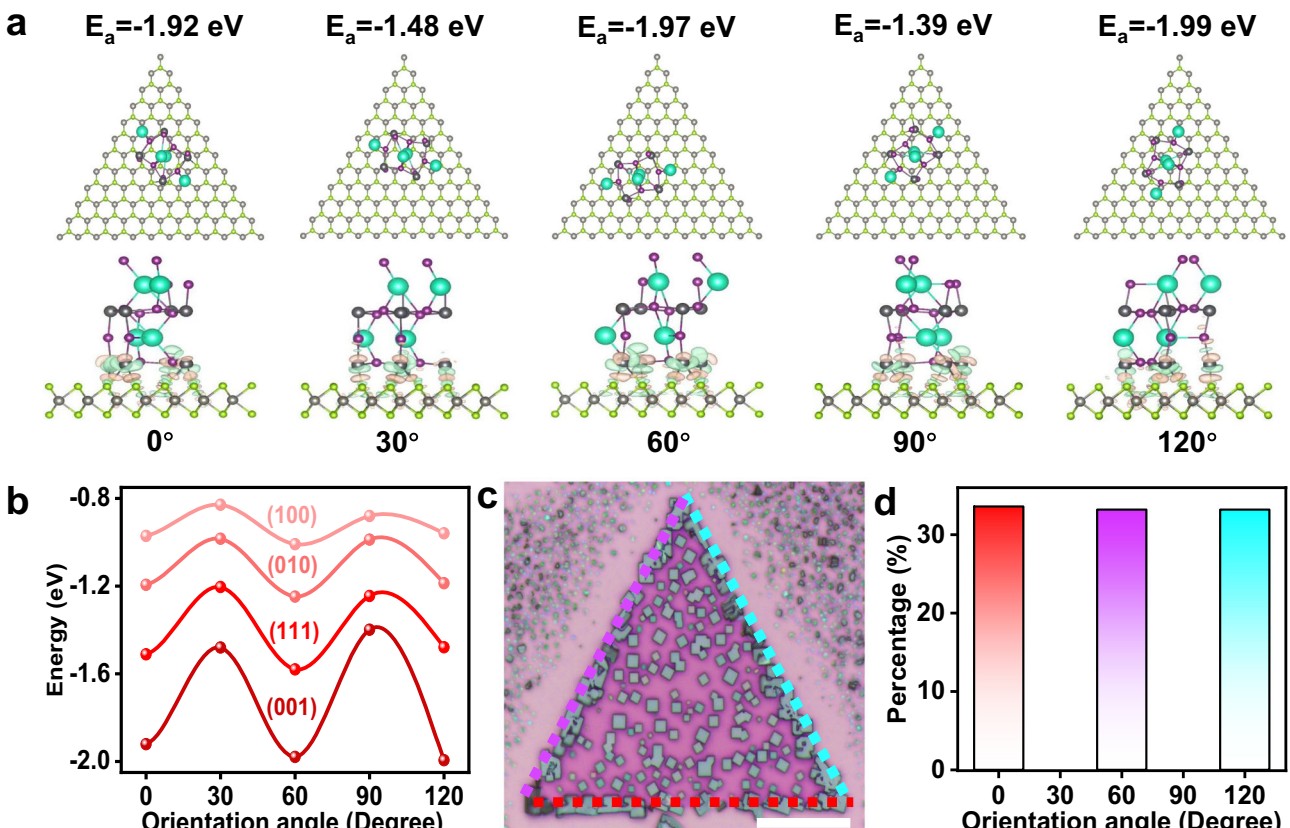

**Fig. 4 | Mechanism of facet-selective epitaxial growth. a** Interfacial structural models of epitaxial CsPbI$_3$ crystals with (002) out-of-plane facet on monolayer WSe$_2$ at possible orientation angles. **b** The calculated formation energy of epitaxial CsPbI$_3$ per unit cell volume with different facets on monolayer WSe$_2$ at possible orientation angles. The symbols represent the calculated formation energy at a given facet and orientation of the epilayer. The solid line represents the energy barrier between different epitaxial orientations. **c** A representative optical image of epitaxial multiple CsPbI$_3$ crystals on single monolayer WSe$_2$. The colored dashed lines represent the epitaxial orientations aligned with the edges of monolayer WSe$_2$. Scale bar, 50 µm. **d** Statistical orientation distribution results of epitaxial CsPbI$_3$ on monolayer WSe$_2$.

demonstrated that the observed result was well-matched with the standard pattern of the orthorhombic phase of CsPbI$_3$ (denoted as CsPbI$_3$ in this manuscript unless otherwise stated) with space group of *Pbnm*. From the XRD patterns, four dominant peaks at 14.6°, 14.8°, 28.6° and 29° (2-theta) are clearly resolved, being indexed to (002), (110), (004) and (220) planes of CsPbI$_3$ and consistent with the previous literature reports. Additionally, the splitting at the (002)/(110) and (004)/(220) peaks is direct evidence that rules out the cubic phase of CsPbI$_3$[35,36]. Furthermore, the relative intensity of (002) and (004) peaks compared with those of splitting peaks become substantially stronger. This fact together with the absence of (020) peak clearly indicate the preferential growth orientation of CsPbI$_3$ along (002) facet shown in Fig. 3e. Detailed discussion is provided in Supplementary Figs. 8, 9.

The completely disparate lattice structures between CsPbI$_3$ perovskite and monolayer WSe$_2$ may easily determine a condition of complete incommensurateness because the exactly conformal lattices with one-to-one bond matching are highly unlikely. However, three azimuthal orientations of CsPbI$_3$ were obviously observed (Fig. 1b) and the [010] alignment of the CsPbI$_3$ perovskite single crystals was primarily aligned to the three equivalent directions of the monolayer WSe$_2$: [$\bar{1}$100], [10$\bar{1}$0] and [0$\bar{1}$10], which strongly implied that the azimuthal orientations in incommensurate epitaxy exist[37]. Combining the above analysis and experimental results, a coincidence site lattice matching model would be proposed to account for the possible epitaxial relationship (Fig. 3f): CsPbI$_3$ propagates along [100] and [010] directions that are paralleled to [$\bar{1}$2$\bar{1}$0] and [10$\bar{1}$0] directions of the

underlying monolayer WSe$_2$, respectively. Corresponding lattice parameter analysis suggests that along CsPbI$_3$ [100] direction, one unit cell of CsPbI$_3$ can well match with three unit cells of WSe$_2$ [$\bar{1}$2$\bar{1}$0], with a −0.03% lattice mismatch; while along CsPbI$_3$ [010] direction, one unit cells of CsPbI$_3$ can well match with five unit cells of WSe$_2$ [10$\bar{1}$0], with a −0.04% lattice mismatch.

From a thermodynamic perspective, the orientation action is initiated from the interfacial layer via determining the arrangement of arriving adatoms, which is typically governed by the difference in interfacial formation energies among the possible crystal facets and orientations. The outcome is a scenario where the energetically favorable facets and orientations dominate the epitaxial growth. We then quantitatively rationalize the epitaxial behaviors by calculating the interfacial formation energies vs. exposed out-of-plane facets and in-plane rotation angles (θ) of CsPbI$_3$ on monolayer WSe$_2$ (0001) using density functional theory (DFT). Due to the threefold and fourfold symmetry of WSe$_2$ (0001) and CsPbI$_3$, respectively, calculation models only need to consider θ ∈ [0°, 120°], and set the possible values to be θ = 0°, 30°, 60°, 90° and 120° for comparative analysis. The basic atomic structures and the in-plane rotation angle definition are presented in Supplementary Fig. 10. Figure 4a presents the interfacial atomic structures when epitaxial CsPbI$_3$ crystals with (002) out-of-plane facet stacked on monolayer WSe$_2$ at possible orientation angles, other interfacial structural models are arranged in Supplementary Fig. 11. Figure 4b summarized the interfacial formation energies of epitaxial CsPbI$_3$ per unit volume with different facets on monolayer WSe$_2$ at possible orientation angles. According to the Minimum Energy

Principle, the stable interfacial atomic configurations are energetically favorable orientations at 0°, 60°, and 120° and with an out-of-plane facet of (002), which are highly consistent with the experimental observations. Quantitatively, the interfacial formation energies per unit cell volume for orientations of 0°, 60°, and 120° are −1.92 eV, −1.97 eV, and −1.99 eV, respectively, all of which are much lower than those for 30° and 90° (−1.48 eV and −1.39 eV, respectively). This may be attributed to the different local dipole alignments stemming from the polarity and the sequential internal charge transfer within the monolayer $WSe_2$ that greatly increases the surface energy inhomogeneity. Therefore, well-defined epitaxy through guiding adatom arrangement on 2D $WSe_2$ surface is driven by the favorable interfacial energy. Specifically, when the incoming adatoms are initially adsorbed at energetically favorable sites for nucleation, the configuration would keep changing to adapt to the global energetic minimum. The minimal energy difference ($\Delta E$) between stable−metastable alignments (for example 0° and 30°) reaches −0.44 eV, and this preferred alignment has a thermodynamic probability of $1/(1 + \exp(\frac{\Delta E}{k_B T})) = 99.8\%$, where $k_B$ is the Boltzmann constant and $T = 793$ K is the growth temperature, demonstrating the robust guiding ability of monolayer $WSe_2$ for $CsPbI_3$ epitaxial growth[38]. This prediction can be well validated by the experimental observation (Fig. 4c) and statistical alignment distribution results shown in Fig. 4d. The probabilities for three stable alignments are almost equal (33.6%, 33.2%, and 33.2%), which are in line with the distribution of the interfacial formation energies. Overall, the van der Waals surface and threefold rotational symmetry lattice structure of the monolayer $WSe_2$ hold the key to breaking the lattice matching rule and are fundamentally responsible for the well-defined epitaxial nucleation and growth, during which the arriving adatoms are directed to form an oriented stable nucleus owing to favorable interfacial energy.

Also, the experimental observations show that epitaxial nucleation density is exponentially decreased, and the growth rate is accelerated with temperature increase. The epitaxial nucleation density and growth dynamics are significantly impacted by the dangling bond-free surface of monolayer $WSe_2$. Specifically, the van der Waals surface can significantly lower the adatom diffusion energy barrier and thus render a much longer surface diffusion distance, so that the relaxation of adatoms to energetically favorable sites is not limited by kinetic barriers during epitaxy[39]. The unique energy landscape of the van der Waals surface essentially modifies the epitaxial nucleation and growth process. More specifically, when increasing the temperature, the thermal energy-driven diffusion length is greatly prolonged, and the long-range migration of adatoms to energetically favorable sites for large critical-size nucleation formation is kinetically favored. Simultaneously, the higher kinetic energy will inevitably give rise to much easier desorption of adatoms. Consequently, simultaneous collisions of arriving adatoms on the inert surface are greatly suppressed, which leads to large nuclei with reduced density. By contrast, at low temperature, the diffusion length driven by lower kinetic energy becomes shorter, and the corresponding desorption probability of adatom is reduced. This scenario thus offers a high probability of collision among diffusing adatoms, thus leading to smaller nuclei with increased quantity. The subsequent in-plane growth shares the same driving force as nucleation via van der Waals forces, which proceeds through diffusion and registry of the adatoms to the growing front of stable nuclei. At high temperature, in addition to the long mean free path of adatomic diffusion, the high local supersaturating gradient is synchronously introduced. Once the stable nuclei are formed, adatoms can easily and quickly transport to the edges for continuously feeding the in-plane epitaxial growth. Due to the sparser nuclei distribution, the single nucleation can receive sufficient nutrient atoms, thus accelerating the growth rate. In the case of lower-temperature growth, the opposite case happens. Quantitative analysis and kinetic mechanism are well organized in Supplementary Figs. 12, 13.

## Minimized disorder in $CsPbI_2Br/WSe_2$ heterostructures

To probe the optical quality of the epitaxial perovskite, we first carried out the steady-state ultraviolet-visible (UV-vis) absorption spectra of $CsPbI_2Br$ nanoplate and $CsPbI_2Br/WSe_2$ heterostructure (Fig. 5a). The $CsPbI_2Br/WSe_2$ heterostructure shows a sharper band-tail state and lower sub-band gap absorption below the optimal band gap as compared to $CsPbI_2Br$ solid, showing a significantly lowered trap state density in epitaxial lattices. The corresponding optical band gap ($E_g$) extracted from Tauc plots displays a red shift (30 meV) in $CsPbI_2Br$ nanoplate, which results from structural disorders and surface defect states (Fig. 5b). These disorders cause local band minima and maxima stemming from local electrostatic potential fluctuations, which form a high density of low-energy band-tail states. The Urbach energy ($E_U$), a critical parameter for quantitatively verifying the degree of energetic disorder based on the Urbach rule: $\alpha = \alpha_0 \exp(E/E_U)$, where $\alpha$ is the absorbance coefficient, $\alpha_0$ is a constant, and $E$ is the excitation energy[40,41], of $CsPbI_2Br/WSe_2$ heterostructure (6.5 meV) is lower than that of $CsPbI_2Br$ solid (10.4 meV), exhibiting a homogeneous and flat energy landscape (Fig. 5c). These results strongly suggest that the van der Waals epitaxial $CsPbI_2Br/WSe_2$ heterostructure is much cleaner than $CsPbI_2Br$ nanoplate so that more photo-generated carriers would contribute to the efficient gain buildup and fast population inversion, leading to low-threshold laser.

To investigate the effect of energetic disorder on energy funneling following light absorption, the spectral distribution and relaxation processes of photoexcited carriers are examined by transient absorption (TA) experiment under low pump fluence ($\sim 1$ μJ cm$^{-2}$ to eliminate the effect of Auger recombination, the pump-probe system is equipped with 400 nm laser pulse as the excitation source and a white-light continuum probe). The spectro-temporal TA maps of $CsPbI_2Br$ nanoplate and $CsPbI_2Br/WSe_2$ heterostructure are shown in Fig. 5d, e. The initial positions in both samples of the bleaching formation peaks below 1.5 ps experience a distinct and fast red-shift (marked by red dotted curves), which originates from the hot carrier-induced competition between the band-gap renormalization effect and Burstein−Moss effect[42]. The transient bleaching peaks of both $CsPbI_2Br$ nanoplate and $CsPbI_2Br/WSe_2$ heterostructure progressively red-shifted during the photo-bleaching recovery, which suggested energy funneling of the photoexcited carriers to low-energy states[43]. A smaller energy shift of 14.3 meV was observed for $CsPbI_2Br/WSe_2$ heterostructure, whereas more than two times larger energy shift of 35.6 meV was observed for $CsPbI_2Br$ nanoplate (Fig. 5f), agreeing well with red-shift amplitude measured by the steady-state absorption spectra, which is caused by the redistribution of excited carrier population via Urbach tails-induced photon reabsorption. This trend, consistent with band-tail sharpening in $CsPbI_2Br/WSe_2$ heterostructure, reflects shallower trap states, lower defect concentration, and homogeneous energy landscape, suggesting suppressed energy funneling events and optimal crystal quality promising for enhanced optical gain. The photoexcited carrier relaxation processes ranging from several hundreds of picoseconds to nanoseconds show a typical bleaching recovery, which originates from the long-range near-surface carrier diffusion in $CsPbI_2Br$ bulk to heterointerface and ensuing transfer. The timescale of lasing onset is far faster than the competing non-radiative recombination brought by carrier quenching of monolayer $WSe_2$, elucidating why the lasing action is impervious to the presence of the monolayer $WSe_2$.

It has been well demonstrated that the defects of monocrystalline halide perovskite are mainly located on the surface, which can have a major impact on optical-electronic properties and device performances[15]. The transient reflection (TR) spectroscopy here is employed to probe the surface disorder landscape of $CsPbI_2Br/WSe_2$ heterostructure and $CsPbI_2Br$ nanoplate and provides deep insights into the surface charge carrier dynamics of each case. The pump penetration depth is calculated as 83 nm from the absorption

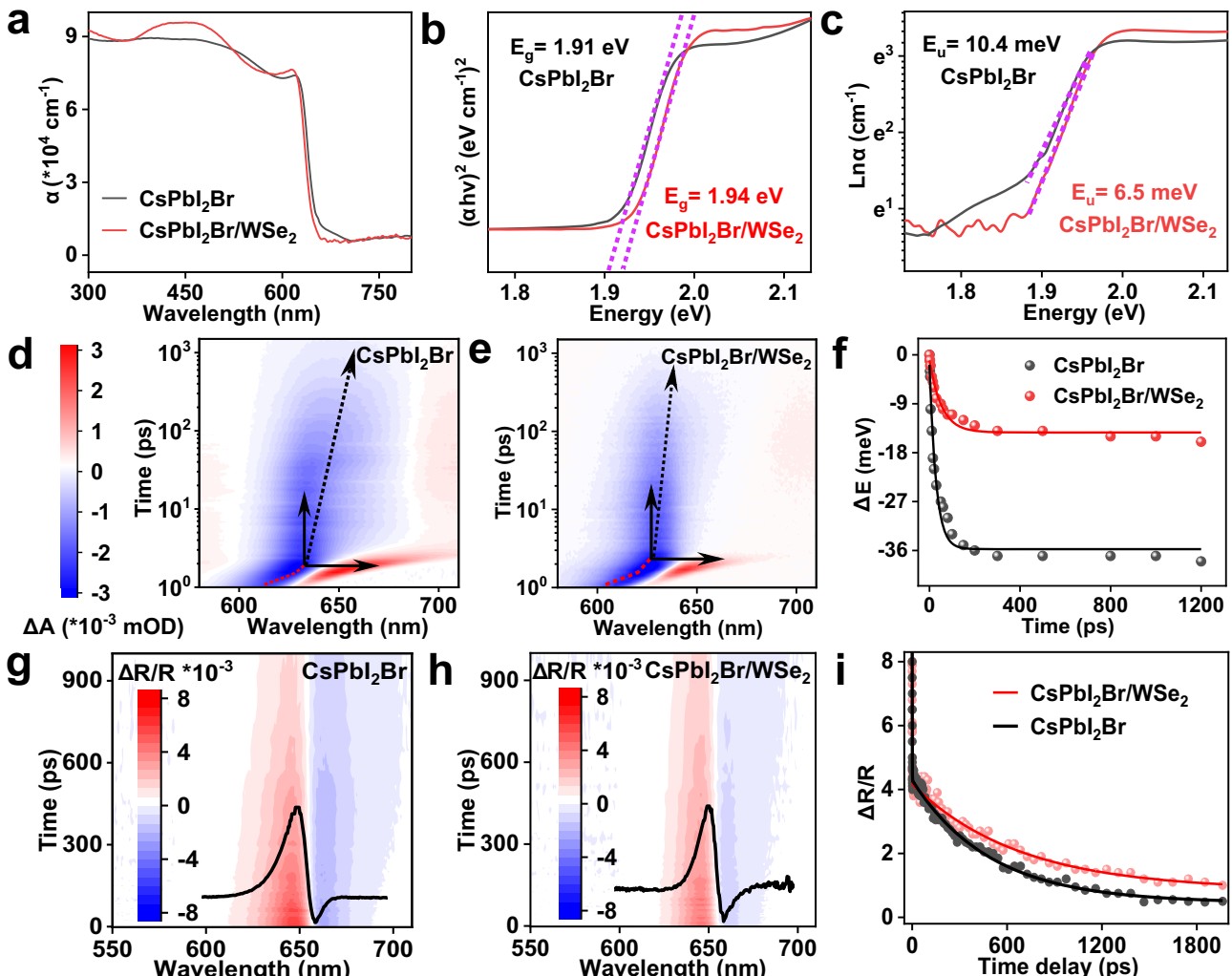

**Fig. 5 | Energetic disorder landscape. a** Steady-state ultraviolet-visible (UV-vis) absorption spectra of CsPbI$_2$Br nanoplate and CsPbI$_2$Br/WSe$_2$ heterostructure. **b** The Tauc plots of absorption spectra for extracting the optical bandgap of CsPbI$_2$Br nanoplate and CsPbI$_2$Br/WSe$_2$ heterostructure. The dashed purple lines are Tauc plots for determining the optical bandgap. **c** Logarithm of absorption coefficient versus photon energy to calculate Urbach energy ($E_U$) of CsPbI$_2$Br nanoplate and CsPbI$_2$Br/WSe$_2$ heterostructure. The dashed purple lines mark the sub-bandgap absorption where Urbach energies are extracted. **d**, **e** Spectro-temporal transient absorption maps for CsPbI$_2$Br nanoplate and CsPbI$_2$Br/WSe$_2$ heterostructure. The dashed arrow is a guide to the eye for highlighting the spectra shift over time. **f** The shifts of peak position of transient bleach over time for CsPbI$_2$Br nanoplate and CsPbI$_2$Br/WSe$_2$ heterostructure. The symbols represent the experimental values, and the solid lines are fits to an exponential decay function. **g**, **h** Spectro-temporal transient reflectance maps for CsPbI$_2$Br nanoplate and CsPbI$_2$Br/WSe$_2$ heterostructure. The black curves are corresponding typical spectra cuts at 5 ps. **i** The surface carrier decay kinetics extracted from the TR spectra of CsPbI$_2$Br/WSe$_2$ heterostructure and CsPbI$_2$Br nanoplate and respective fitting to the diffusion-surface recombination model. The red dots and black dots represent experimental data, and the red line and black line correspond to their fitting results.

coefficient while the effective detecting depth is about 20 nm for probe photon energy near bandgap based on $\lambda/4\pi n(\lambda$ is the central wavelength, $n$ is refractive index)[44,45]. The prototypical 2D pseudocolor images of the TR spectra under low excitation fluence are shown in Fig. 5g, h for CsPbI$_2$Br nanoplate and CsPbI$_2$Br/WSe$_2$ heterostructure, respectively. Anti-symmetric peaks signifying the bleach of the charge carrier transition dominate the TR profiles, which is caused by phase-space filling in the presence of free photocarriers. The surface carrier dynamics, including defect-triggered surface recombination and carrier diffusion from the surface into the bulk, is described by the corresponding reflectance change ($\Delta R/R$) kinetics (Fig. 5i). Due to low carrier concentration gradient created by weak pump fluence, the surface carrier depopulation is mainly contributed by defect-induced surface recombination. The decay kinetic results show a very sluggish decay for CsPbI$_2$Br/WSe$_2$ heterostructure (674.7 ps) compared to that of CsPbI$_2$Br nanoplate (495.3 ps), indicating the retardative surface trap-assisted recombination. Based on the diffusion-surface

recombination model (Supplementary Note 1)[44–46], the carrier diffusion coefficient ($D$) and the surface recombination velocity ($S$) are extracted to be 0.52 cm$^2$ s$^{-1}$ and 1300 cm s$^{-1}$ for CsPbI$_2$Br/WSe$_2$ van der Waals heterostructure and 0.43 cm$^2$ s$^{-1}$ and 3700 cm s$^{-1}$ for CsPbI$_2$Br nanoplate. The results robustly substantiate the reduced defect concentration on CsPbI$_2$Br/WSe$_2$ surface. From these data, the surface defect density is quantitatively estimated based on the following equation: $S = \sigma \nu_{th} N_t$, where $\nu_{th} \approx 3.7 \times 10^{10}$ cm s$^{-1}$ is the carrier thermal velocity, $\sigma \approx 10^{-15}$ cm$^2$ is a typical recombination surface cross section in semiconductor, $N_t$ is the number of recombination centers per square centimeter[47]. Under these assumptions, we deduced the surface defect density to be $3.5 \times 10^{10}$ cm$^{-2}$ for CsPbI$_2$Br/WSe$_2$ heterostructure, which is three times lower than that of CsPbI$_2$Br nanoplate ($1 \times 10^{11}$ cm$^{-2}$).

To characterize the spatial crystallinity of epitaxial perovskite, the cross-sectional high angle annular dark field scanning transmission electron microscope ((HAADF-STEM)) was performed, the detailed information is presented in Supplementary Fig. 14. The corresponding

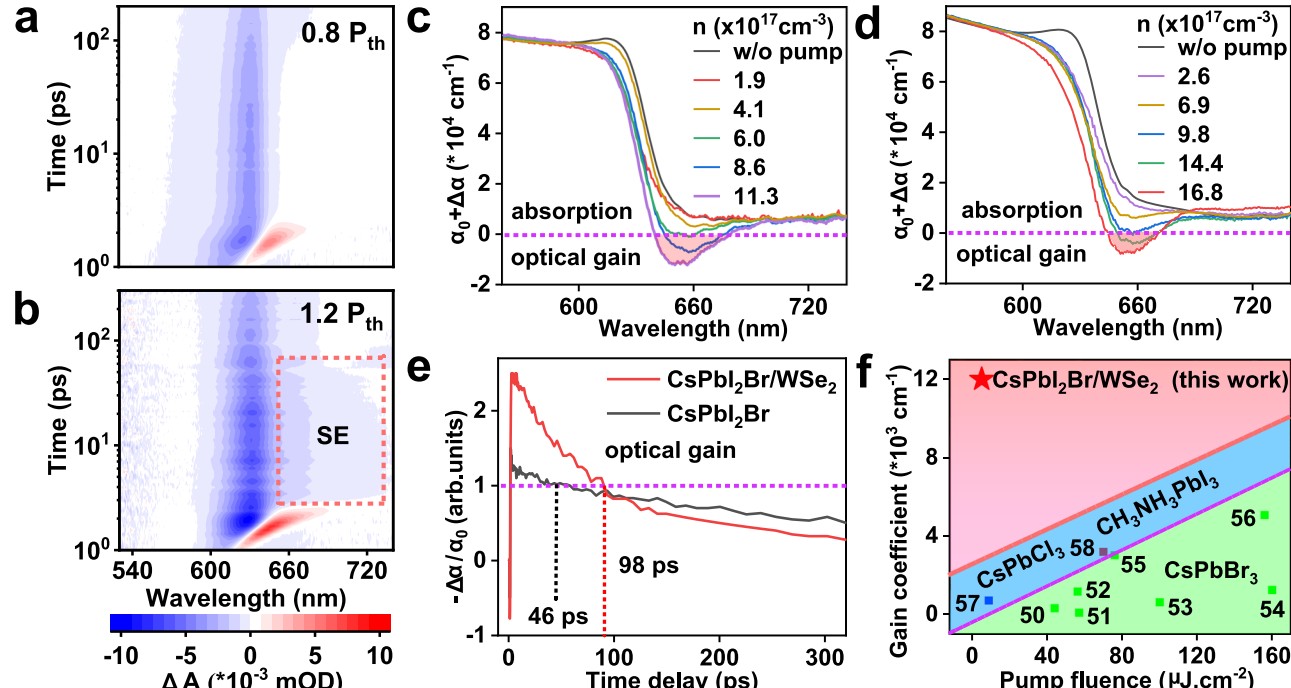

**Fig. 6 | Optical-gain responses. a** Transient absorption (TA) spectroscopic data of $CsPbI_2Br/WSe_2$ heterostructure with photoexcitation at 0.8 $P_{th}$, $P_{th}$ represents the corresponding lasing threshold. **b** The stimulated emission (SE) feature in TA map, is manifested as a pronounced bleach signal under 1.2 $P_{th}$ fluence. **c, d** Carrier density-dependent nonlinear absorption spectra of $CsPbI_2Br/WSe_2$ heterostructure and $CsPbI_2Br$ nanoplate, respectively, which were recorded at a delay time of ~3 ps. **e** TA kinetics plots of the SE dynamics at 660 nm (in resonance with the lasing peak) in the form of $-\Delta\alpha/\alpha_0$ versus time delay. The $\alpha$ and $\alpha_0$ are the absorption

coefficients under pumped and unpumped conditions, respectively. The gain decay lifetime corresponds to $-\Delta\alpha/\alpha_0 > 1$, demonstrating a prolonged gain lifetime in $CsPbI_2Br/WSe_2$ heterostructure. **f** Comparison of representative optical gain values reported for perovskite-based materials[50-58]. The green and blue shaded areas represent the reported gain coefficients, and the red shaded area highlights our result, which belongs to the premier echelon. The dashed lines classify the reported perovskite gain materials, most of which focus on the green ($CsPbBr_3$) emitters.

selected area electron diffraction (SAED) pattern shows a single set of separated diffraction spots, strongly confirming that the epitaxial perovskite belongs to a single crystal. Accordingly, the atomic-resolution HAADF-STEM observations show that the epitaxial perovskite lattices are free from defects and vacancies over a large area, demonstrating very high crystallinity quality. The zoomed-in image reveals a nearly perfect periodic atom arrangement. The clearly resolved lattice spacing of 0.58 nm corresponds to the (001) plane of perovskite crystal, which is well consistent with the SAED result. Furthermore, the corresponding cross-sectional EDS elemental mapping images clearly display the spatial distribution of the constituent elements, whose uniform color contrasts further confirm the compositional homogeneity. These direct cross-sectional images convincingly prove the high crystallinity quality of our heterostructures, which leads to high structural ordering and minimized energetic disorder.

## Optical-gain responses of $CsPbI_2Br/WSe_2$ heterostructures

Lasing oscillation can emanate from an optically or electrically pumped semiconductor gain medium when round-trip gains overcome round-trip losses to achieve the population inversion state in a resonator. Halide perovskites are typically characterized by approximately exponential Urbach tail states below the nominal band edge and abundant surface defects. These energetic disorders reduce the optical gain achieved by a given injected carrier density and consequently impact the lasing behaviors, such as the lasing threshold. To better understand the optical-gain responses, three gain metrics are used for benchmarking: the material gain coefficient, the threshold carrier density, and the gain lifetime, which can be realistically probed by femtosecond TA spectroscopy[48,49]. Figure 6a, b show representative 2D TA maps for $CsPbI_2Br/WSe_2$ heterostructure at different pump intensities (0.8 $P_{th}$ and 1.2 $P_{th}$, $P_{th}$ represents the lasing threshold) to

track the lasing dynamics. These 2D maps mainly consist of a distinct ground-state bleaching (GSB) band and a broad short-lived photo-induced absorption band located at the low-energy side arising from band gap renormalization, which is the direct indicator of the dynamic transition of hot carriers. When the pump intensity increases above the lasing threshold, the profound photoinduced bleaching feature at 650 – 700 nm from stimulated emission (SE) occurs. The spectral position of the SE peak confirms that the onset of SE is closely followed by the hot carrier cooling, suggesting that photoexcited carriers must thermally relax to the band edge to contribute to population inversion. The kinetic of that extracted at 660 nm consists of a rapid buildup time of 0.76 ps (on the timescale of hot carrier cooling) and an optical gain decay lifetime of 86 ps (Supplementary Fig. 15). Specifically, with high-intensity excitation, the electrons are pumped to the hot states in conduction band at first and then they relax promptly from the nonequilibrium hot states to the emissive states within ~0.76 ps, which is far faster than the gain decay process. Therefore, the population in the emissive states can be accumulated to feed the population inversion within its lifetime.

The hallmark of optical gain is photo-induced bleaching ($\Delta\alpha = \alpha - \alpha_0 < 0$, where $\alpha$ and $\alpha_0$ are the absorption coefficients under pumped and unpumped conditions, respectively), whose magnitude is greater than $\alpha_0$ ($-\Delta\alpha/\alpha_0 > 1$)[48,49]. Figure 6c, d display dynamic gain development of $CsPbI_2Br/WSe_2$ heterostructure and $CsPbI_2Br$ nanoplate based on spectral cuts of the transient intrinsic absorption coefficient $\alpha = \alpha_0 + \Delta\alpha$ at 3 ps under different carrier density. The $\alpha$ in both samples experience a pronounced reduction of the band edge absorbance and the slight increase of the short-wavelength absorbance, while on the long wavelength side, a region where $\alpha$ becomes negative is tantamount to optical gain (G). The calculated G based on $G = -\alpha$ are 8296 $cm^{-1}$ for $CsPbI_2Br$ nanoplate and 12,000 $cm^{-1}$ for

CsPbI$_2$Br/WSe$_2$ heterostructure, which outperforms most of the state-of-art reports, as shown in Fig. 6f [50–58]. The decay lifetimes, defined as the time interval in which $-\Delta\alpha/\alpha_0$ maintains a value higher than 1, are 46 ps for CsPbI$_2$Br nanoplate and 98 ps for CsPbI$_2$Br/WSe$_2$ heterostructure (Fig. 6e), indicating suppressed nonradiative Auger recombination at high pump fluence. Transparency condition (zero net absorption) is achieved at a threshold carrier density of $6.0 \times 10^{17}$ cm$^{-3}$ for CsPbI$_2$Br/WSe$_2$ heterostructure, which is lower than that of CsPbI$_2$Br nanoplate ($9.8 \times 10^{17}$ cm$^{-3}$), and is beneficial to form a large gain cross-section in the lasing band. Indeed, the comprehensively enhanced optical gain properties not only demonstrate the advantage of flattened energetic disorder landscape in perovskite lattices by van der Waals epitaxy on monolayer semiconductors, but also are critical for enabling low-threshold lasing.

## Reduced laser threshold of CsPbI$_2$Br/WSe$_2$ heterostructures

The above-mentioned investigations have shown that our epitaxial heterostructures have combined unique properties for the photonic lasing application. Firstly, the epitaxial monocrystalline halide perovskites naturally yield a well-defined photonic micro-cavity geometry with whispering gallery mode, which can simultaneously function as lasing gain media and optical feedback supplier to support optical amplification through stimulated emission within themselves. More importantly, the epitaxial halide perovskite/2D semiconductor heterostructures are characterized by minimized energetic disorder and enhanced optical gain properties compared with individual perovskite crystals. Secondly, the tunable 2D semiconductor layers cleanly interfaced with perovskites in heterostructures can serve as charge carrier injection layers, which offer potential ready-to-use material options for developing future perovskite-based electronically driven lasers. Thirdly, the van der Waals epitaxial halide perovskite/2D semiconductor heterostructures are able to monolithically integrate on both CMOS-compatible substrate (SiO$_2$/Si) and photonic-compatible platforms (Si and LiNbO$_3$). This general applicability offers a diverse and versatile material platform for designing on-chip reconfigurable photonic-electronic integrated devices. Therefore, these advantages and promises motivate us to explore the photonic lasing application of epitaxial heterostructures.

As shown in Fig. 7a, b, 3D-view atomic force microscopy (AFM) and top-view scanning electron microscope (SEM) of heterostructures display that the epitaxial perovskite is a perfectly quadrated single crystal. It thus not only functions as an active optical gain medium but also serves as a self-organized optical resonator for providing effective optical feedback to support lasing action. The atomically smooth surface and flat end facets are beneficial to lower the lasing threshold because it would well reduce the scattering loss, improve the light confinement, and facilitate effective optical resonance. Figure 7c, d show PL/lasing images of individual CsPbI$_2$Br/WSe$_2$ heterostructure below and above the lasing threshold, respectively. Below the lasing threshold, the whole nanoplate body exhibits minimal color contrast, suggesting the unidirectional out-coupling of spontaneous photoluminescence emission. In comparison, the image of CsPbI$_2$B/WSe$_2$ heterostructure excited above the threshold shows bright red emissions localized at the four corners and weak PL at the body due to selectively spatial interference of the coherent light sources, clearly demonstrating the existence of whispering gallery mode (WGM) lasing mode. Indeed, the WGM microcavity could sustain a traveling-wave optical field in the 2D plane, and thus the emission would be confined by total internal multi-reflection at cavity boundaries due to the large difference in refractive index between the perovskite resonator and its surrounding environment, marked by red dotted lines in Fig. 7b.

To show the influence of the heterostructure on the lasing behaviors, the pump fluence-dependent PL spectra and intensity/FWHM plots of CsPbI$_2$Br nanoplate (Fig. 7e, h), CsPbI$_2$Br/WSe$_2$ heterostructure (Fig. 7f, i) and CsPbI$_2$Br/WS$_2$ heterostructure (Fig. 7g, j) with

comparable thickness of 175–190 nm and edge length of 10–12 μm are summarized. In each case at a low pump level, the PL spectrum is broad and featureless (Gaussian shape with full width at half maximum (FWHM) ~32–35 nm); as the pump power is increased near the lasing threshold, two adjacent red-shift PL shoulders begin to appear near the tail of the absorption edge; with a continuous increment of pump intensity, the neighboring PL spectra merge into single sharp PL peak (Lorentz shape with FWHM~ 0.7–1.0 nm), the integrated intensity synchronously experiences a sharply nonlinear growth. The corresponding light-in–light-out (L–L) data and FWHM plots as a function of pump fluence show a distinct S-shape profile, suggesting a distinct transition from PL to lasing and then saturation. This result indicates that the spatially neighboring seed photons within the volume excitation can participate in seeding the avalanchestimulated emission to trigger lasing oscillation, with spontaneous emission coupling factor $\beta$ estimated to be 0.05 (Supplementary Note 2)[59,60]. Furthermore, the non-linear growth in the spectrally integrated intensity and a simultaneous FWHM narrowing are preliminary evidences of lasing action, where the threshold can be extracted to be 1.98 μJ cm$^{-2}$ for CsPbI$_2$Br/WSe$_2$ heterostructure and 2.21 μJ cm$^{-2}$ for CsPbI$_2$Br/WS$_2$ heterostructure, which all are much lower than that of CsPbI$_2$Br nanoplate grown on SiO$_2$/Si substrate (6.36 μJ cm$^{-2}$). By comparison, the obtained lasing threshold of CsPbI$_2$Br/WSe$_2$ heterostructure is also much lower than previous reports (Supplementary Table 2), confirming the advantage of our epitaxial CsPbI$_2$Br/WSe$_2$ heterostructure for lasing. The coherence property is comprehensively assessed by mode narrowing and output polarization, which are inherently associated with temporally and spatially coherent lasing emission[61–63]. The detailed information is presented in Supplementary Fig. 16. The spectra linewidth is sharply narrowed down to 0.7 nm (<1.0 nm) with above-threshold excitation, which is well consistent with the emission characteristic of halide perovskite laser and with the Schawlow–Townes equations[62,64]. This clearly signifies the emission transition from incoherent to coherent[65]. The estimated coherent length is as long as 622 μm, which is longer than those recently reported halide perovskite-based lasers[66,67]. The corresponding coherent time is 0.21 ps, strongly suggesting the coherent emission[68]. Furthermore, the degree of polarization of the lasing mode was measured up to 69%, indicating robust polarization selectivity and good spatial coherence[69]. This may originate from the competition between three-fold degenerate bright-triplet and dark singlet transitions and optical birefringence from Vernier-effect coupling in self-organized perovskite laser[66,70]. The obvious threshold behavior, sharp linewidth narrowing, monochromatic lasing output, long-range coherence, high-bright lasing output beams, and a large degree of emission polarization unambiguously demonstrate the onset of lasing action. Of the CsPbI$_2$Br/WSe$_2$ heterostructure examined with perovskite thickness spanning from 170–200 nm and edge length ranging from 9 μm to 18 μm (36 in total), more than 85% showed single-mode lasing with narrow lasing threshold distribution ranging from 1.95 μJ cm$^{-2}$ to 3.87 μJ cm$^{-2}$. However, only 47% CsPbI$_2$Br nanoplates grown on SiO$_2$/Si substrate were able to lase and showed scattering lasing threshold distribution spanning from 6.36 μJ cm$^{-2}$ to 23.89 μJ cm$^{-2}$ (Supplementary Fig. 17). This statistical analysis clearly demonstrated that the van der Waals epitaxial CsPbI$_2$Br/WSe$_2$ heterostructure laser with single lasing mode is highly reproducible. This huge discrepancy may be derived from the inhomogeneous energy landscape, where a high degree of energy disorder widens the distribution of electronic states and creates band-tail states. These abundant localized disorder states inevitably lead to severe extra energy loss, less centralized excited energy levels, and suppressed radiation transition, which increases the threshold of population inversion and is unfavorable for light amplification.

The experimentally observed red-shift feature of the lasing peak compared to the normal PL peak and corresponding threshold carrier density of $\rho_{th} = 6.1 \times 10^{17}$ cm$^{-3}$, which is higher than the Mott density of

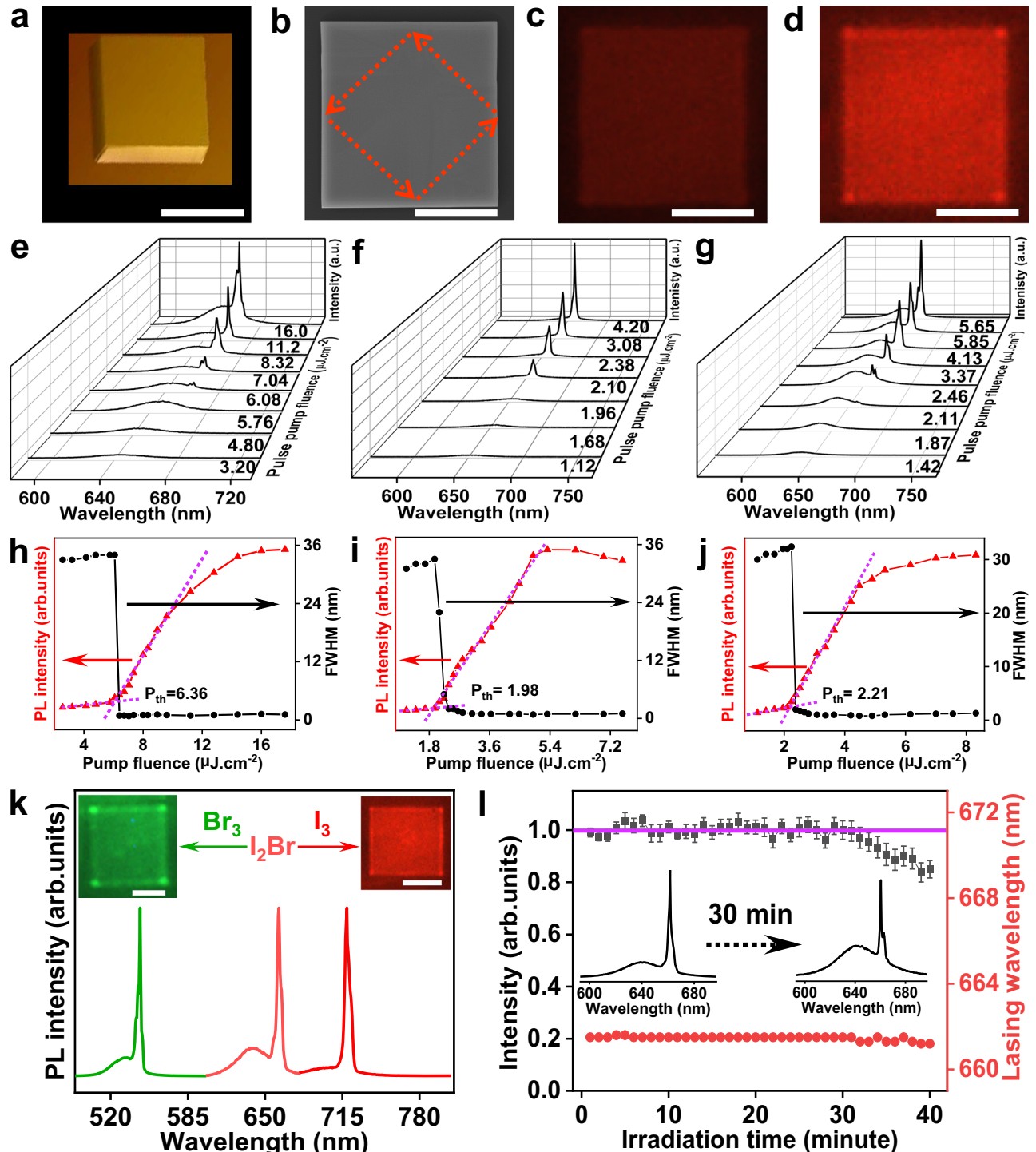

**Fig. 7 | Enhanced lasing ability of halide perovskite/2D semiconductor heterostructures. a, b** 3D AFM and SEM images of CsPbI$_2$Br perovskite epitaxially grown on monolayer WSe$_2$ with 183 nm thickness, showing smooth surface and sharp edges for the high-quality optical resonator. The red dotted arrow indicates the light propagation inside the square whispering gallery mode (WGM) microcavity. **c, d** PL emission images below and above the lasing threshold for CsPbI$_2$Br/WSe$_2$ heterostructure, respectively. Scale bars, 10 μm. The pump fluence-dependent PL spectra and intensity/FWHM plots of CsPbI$_2$Br (**e, h**) and CsPbI$_2$Br/WSe$_2$ (**f, i**) and CsPbI$_2$Br/WS$_2$ (**g, j**). The purple dashed lines represent the fits to the experimental data, showing the threshold region. Due to the limited space, the a.u.

represents the arb. units. **k** Color-tunable lasing spectra and their images of epitaxial perovskite CsPbX$_3$ on monolayer WSe$_2$ with different halide ion species. Spectra from left to right show the green and red lasing output, corresponding to CsPbBr$_3$, CsPbI$_2$Br, and CsPbI$_3$ gain mediums. Insets display the corresponding lasing emission images above the threshold. Scale bars, 5 μm. **l** Photo-stability of lasing action assessed by monitoring the lasing intensity and lasing wavelength as a function of laser irradiation time at 1 kHz repetition rate in the ambient environment. The error bars are recorded by standard deviation. Insets are respective lasing intensity spectra before and after 30 min photo-irradiation.

CsPbI$_2$Br ($2.6 \times 10^{17}$ cm$^{-3}$), clearly suggest that the electron-hole plasma mechanism is primarily responsible for the lasing action of CsPbI$_2$Br/WSe$_2$ heterostructure at room temperature[60,71]. Besides, the red-shifted lasing peak for CsPbI$_2$Br/WSe$_2$ heterostructure centered at 658.4 nm, the position of which for CsPbI$_2$Br nanoplate is further shifted to 663.8 nm. This observation can be understood on the basis of the fundamental absorption rule of semiconductors and the Urbach rule, more details are presented in Supplementary Note 3. Specifically, the optical gain spectrum or the lasing regime is generally governed by the balance between stimulated emission and absorption processes. As the stimulated emission transports along optical waveguide paths inside microcavity, the photon reabsorption and reemission processes proceed repeatedly due to the presence of the Urbach tail. The ree-mitted photon typically has a longer wavelength than that of the reabsorbed photon due to the inevitable energy loss of the carriers. Consequently, the optical gain will shift to a longer wavelength as the band tails broaden and the energetic disorder increases, eventually inducing the additional redshift of the lasing peak[72]. The maximum quality factors ($Q = \lambda/\delta\lambda$, where $\lambda$ is the central peak wavelength and $\delta\lambda$ is the peak linewidth) of the monocrystalline lasing cavity are calculated to be 1103 and 827 for CsPbI$_2$Br/WSe$_2$ heterostructure and CsPbI$_2$Br nanoplate, respectively. Under high excitation, a small blue shift, broadening of the lasing central peak, and the emergence of a tiny side-peak within the lasing oscillation peak are observed. These phenomena are the results of gain fluctuation from carrier screening-induced group index change, many-body interactions, and spatial hole burning at high carrier density.

The laser color of the halide perovskite/2D semiconductor heterostructures can be tuned through compositional engineering. Figure 7k demonstrates the room-temperature lasing operation of epitaxial halide perovskites from near-infrared to green, showing bright promise for high-capacity wavelength division multiplexing applications. The photostability of halide perovskite/2D semiconductor heterostructures was assessed by monitoring the lasing intensity as a function of time under laser irradiation (1 kHz repetition rate and 1.5 P$_{th}$) at ambient atmosphere ($25 \pm 1\,°C$, $58 \pm 2\,\%$ relative humidity). Figure 7l shows the variation in lasing intensity of CsPbI$_2$Br/WSe$_2$ heterostructure laser (195 nm thickness and 17 µm edge length), with a standard deviation of 0.26% within ~30 min of continuous irradiation, followed by a 20% drop in output intensity. This is more stable than CsPbI$_2$Br nanoplate grown on SiO$_2$/Si substrate with comparable thickness and edge length (average stability time is about 10 min, see Supplementary Fig. 18a). Additionally, the lasing wavelength of the mixed halide perovskite CsPbI$_2$Br/WSe$_2$ heterostructure maintained unchanged at 661.7 nm, showing no phase segregation occurred under continuous photoexcitation. The excellent optical stability of epitaxial perovskite/2D semiconductor heterostructure is possibly contributed by reduced surface defects and energetic disorder. Furthermore, to enhance the long-term stability of CsPbI$_2$Br/WSe$_2$ heterostructure, large-scale on-chip microprocessor-compatible atomic layer deposition Al$_2$O$_3$ (10 nm) was employed to conformally encapsulate the lasing media, and the stability improved up to 58 days without obvious performance degradation (Supplementary Fig. 18).

To verify the generality of van der Waals halide perovskite/2D semiconductor heterostructures for threshold-reduced lasing application, we comparatively examined the lasing behaviors of CsPbI$_3$/WSe$_2$ (WS$_2$) heterostructures and CsPbI$_3$ nanoplate (Supplementary Fig. 19). The resulting lasing action shared the similar trend with that of CsPbI$_2$Br-based lasers. The lasing thresholds of CsPbI$_3$/WSe$_2$ and CsPbI$_3$/WS$_2$ heterostructures are 11.9 µJ cm$^{-2}$ and 16.8 µJ cm$^{-2}$, respectively, which all are far lower than that of CsPbI$_3$ nanoplate (56.3 µJ cm$^{-2}$). The redshift magnitude of the lasing peak in CsPbI$_3$/WSe$_2$ heterostructure (718.3 nm) is smaller than that of CsPbI$_3$ nanoplate (722.6 nm). These results robustly demonstrate that the improved energetic disorder landscape enabled by van der Waals epitaxy is also responsible for the threshold-reduced lasing action in CsPbI$_3$/WSe$_2$ heterostructure. As for the relatively increased lasing threshold of CsPbI$_3$/WSe$_2$ heterostructure compared with CsPbI$_2$Br/WSe$_2$ heterostructure, the origins may be rooted in reduced Coulombic screening effect due to lower exciton binding energy and hot carrier thermalization-limited optical gain of CsPbI$_3$[73]. Overall, these results critically prove the reliable generality and advantage of our van der Waals epitaxial halide perovskite/2D semiconductor heterostructures for photonic lasing application.

In summary, we demonstrated a straightforward, deterministic bottom-up photonic hetero-integration approach to create a broad range of high-quality and single-crystal halide perovskite/2D semiconductor heterostructures. The epitaxial heterostructures can monolithically integrate on both CMOS-compatible substrates (SiO$_2$/Si) and ready-to-use photonic platforms (Si and LiNbO$_3$). The CsPbI$_2$Br/WSe$_2$ heterostructure shows a large material gain coefficient of 12,000 cm$^{-1}$, ultralow gain threshold, and lengthened gain lifetime due to reduced energetic disorder landscape, which enables ultralow threshold and stable single-mode lasers. The introduced monolayer semiconductor not only endows a robustly facet-selective epitaxial growth of perovskite with well-defined planar configuration required for electronic injection scheme, but also can function as a transport layer required in electrically pumped perovskite lasers, providing a useful all-in-one pathway towards developing electronically pumped on-chip lasers for Si photonics. Furthermore, the artificially high-quality halide perovskite/2D semiconductor heterostructures establish a robust material platform to probe the intrinsic optoelectronic and photonic properties and to design high-performance optoelectronic, photonic, and quantum devices.

## Methods

### Materials and precursors

The materials and precursors used in this study were used as purchased without further purification, which included tungsten selenide power (WSe$_2$, 99.99%, Alfa Aesar), tungsten disulfide powder (WS$_2$, 99.99%, Alfa Aesar), sulfur powder (S, 99.9%, Alfa Aesar), selenium powder (Se, 99%, Sigma-Aldrich), molybdenum trioxide powder (MoO$_3$, 99.9%, Sigma-Aldrich), tin dioxide powder (SnO$_2$, 99%, Macklin). Lead iodide powder (PbI$_2$, 99.99%), cesium iodide powder (CsI, 99.99%), cesium bromide powder (CsBr, 99.99%), Methylamine hydroiodide (MA·HI, 99.99%) all are purchased from Xi'an e-Light New Material Co., Ltd. Argon gas (Ar, 99.99%, Changsha Saizhong Special Gas Co., Ltd.) was used as the carrier gas.

### Synthesis of monolayer WSe$_2$ and WS$_2$

Two-dimension (2D) WSe$_2$ and WS$_2$ semiconductors were synthesized from thermally evaporated vapor-phase reactants directly using bulk powder materials as solid sources in a home-built chemical vapor deposition (CVD) system at atmospheric pressure[74]. The CVD system is uniquely characterized by a switchable gas direction control module, where both sides of the quartz tube are equipped with a gas inlet and outlet controller, enabling the flowing direction of argon gas to be toggled by using the angle-style valves at the two ends of the quartz tube. In the working furnace, a reverse-flowing gas from the growth substrate to the source material is introduced during the temperature-rising process to prevent an undesired supply of vapor source reactant or unintentional nucleation and growth. Upon reaching the pre-defined growth temperature, a forward flow from the source materials to the growth substrate is switched to transport the vapor phase reactant onto the growth substrate for feeding the 2D crystal nucleation and growth. In terms of the experimental procedure, an alumina boat loaded with WSe$_2$ or WS$_2$ powder was firstly placed into the heating zone of a one-inch quartz tube, after which the growth substrate, Si wafer with amorphous silica oxidation on a surface placed on the top of another alumina boat, was located in the downstream region

of the tube, which is 17 cm to 19 cm away from the tube center. The system was flushed with high-purity argon gas at a rate of 632 sccm (standard cubic centimeter per minute) for 10 min to purge the whole tube system to eliminate oxygen and moisture within the tube before material growth. In the growth process, the heating zone was set to ramp from RT (room temperature) to 1180 °C at a rate of 30 °C min$^{-1}$ under ambient pressure, during which an 80 sccm reverse flow of argon gas from the growth substrate to the source was introduced to avoid unintentional nucleation before reaching the predefined nucleation and growth temperature. When the system reached 1180 °C, the gas flow direction was turned to the forward direction from the gas to the growth substrate. The vapor-phase reactants were transported by the flowing argon gas to the downstream deposition region (about 900 °C), where $WSe_2$ or $WS_2$ vapor initiated nucleation and growth on the substrate, and the system was then maintained for 2 min to achieve the desirable 2D crystal growth. After growth, the argon flow direction was immediately reversed again and the furnace was shut down and cooled down naturally.

### Synthesis of monolayer $MoS_2$ and $MoSe_2$

$MoS_2$ and $MoSe_2$ monolayers were grown by a typical CVD route. This tube furnace system established form the conventional CVD structure, which consists of a reaction chamber (quartz tube), gas controller (a gas inlet and outlet channel located at the upstream and downstream, respectively.). The main difference in this system used here is the substrate location, which is put at the top of a source, instead of at a separated position at downstream side of tube, aiming at producing a high supersaturation of vapor-phase precursor environment for $MoS_2$ or $MoSe_2$ growth. An alumina boat containing ~3 g of sulfur or selenium powder was located at the upstream side of the tube. The growth substrates, $SiO_2$/Si, were placed facedown above an alumina boat containing ~0.2 g of $MoO_3$ powder, which was then put into the center of the quartz tube. After removing the oxygen and moisture in the tube by carrier gas of high-purity argon gas with a flowing rate of 635 sccm for 10 min, the reaction was performed at 800 °C (with a ramping rate of 30 °C min$^{-1}$) using argon gas flow of 100 sccm as reactant carrier gas for 5 min, feeding the growth of the 2D $MoS_2$ or $MoSe_2$ crystals. In the case of $MoSe_2$ crystal growth, the carrier gases consist of 200 sccm argon gas and 10 sccm hydrogen gas. After growth, the furnace was shut down and cooled naturally to RT.

### Synthesis of $SnS_2$/$WSe_2$ heterostructures

The $SnS_2$/$WSe_2$ heterostructures were prepared by van der Waals epitaxial growth of $SnS_2$ on a pre-grown $WSe_2$ monolayer[75]. First, $WSe_2$ monolayer on $SiO_2$/Si substrate was prepared by the procedures described above. Next, a quartz boat with S powder (0.2 g) was placed in the upstream zone of the furnace. Another quartz boat with $SnO_2$ powder (0.05 g) was located in the center of the furnace. The monolayer $WSe_2$ on $SiO_2$/Si substrate was tilted above the $SnO_2$ powder. After purging the system with 656 sccm Ar for 10 min, the $SnO_2$ powder was heated to 570 °C (with the temperature of the S powder reaching roughly 180 °C) for 15 min and maintained at that temperature for 7 min under a forward flow of 80 sccm Ar gas. Finally, the reaction chamber was naturally cooled to RT.

### Synthesis of halide perovskite solid sources

All inorganic perovskite sources were synthesized from a solvent-free mechanochemistry process. For $CsPbI_3$ synthesis, a total 14.42 g of stoichiometric mixture of CsI (5.20 g) and $PbI_2$ (9.22 g) were added to a mortar, which were uniformly mixed and milled by pestle for at least 30 min. The solid was then transferred to tube furnace system with argon gas-filled quartz tube at 410 °C and the temperature was kept for 24 h to ensure the solid-state reaction of the two precursors that was completed to form $CsPbI_3$ powder. It is interesting to note that during the natural cooling stage the color of resulting powder is black at

higher temperature (>150 °C), but the color became yellow at RT, which suggests that the formed $CsPbI_3$ powder experienced an thermodynamic-driven phase transition, from photoactive black-phase $CsPbI_3$ to yellow-phase $CsPbI_3$ non-perovskite. Due to reconfigurable and ionic soft latttice nature of metal halide perovskites, such dynamic behavior enables reversible chemical and structural transformation upon exposure to external stimuli such as temperature, which thus offer a unique access to grow photoactive perovskites via vapor deposition process. In the case of $CsPbI_2Br$ perovskite, the overall process shares the similar procedure, expect that the source material fabrication is conducted at a temperature of 400 °C with a stoichiometric recipe of CsBr (4.25 g) and $PbI_2$ (9.22 g).

### Synthesis of all-inorganic halide perovskite/2D semiconductor heterostructures

All-inorganic halide perovskites/2D semiconductor heterostructures were fabricated by vapor phase epitaxial growth protocol. Compared with the system with commonplace CVD system, a pressure control system is introduced, where a mechanical pump is linked into the gas channel downstream for the reduction of chamber inner pressure and a needle valve is equipped at the downstream side of the system to precisely control the reactor pressure. The added pressure controller is designed to adjust the pressure of the reaction chamber and thus manipulate the mass transport dynamics in the perovskite growth process. For synthesis details, an alumina boat containing the $CsPbI_3$ powder source (0.3 g) was put at the heating center of a 1-inch-diameter quartz tube, and the as-grown $WSe_2$ monolayers ($WSe_2$/$SiO_2$/Si) as templates for the growth of $CsPbI_3$ were loaded on anther alumina boat and subsequently inserted into the downstream side of the tube. The growth chamber was pumped down and filled with high-purity argon gas with a flowing rate of 635 sccm for 10 min to maximally remove oxygen and moisture in the tube before being maintained at 200 mbar with 100 sccm of argon gas as the carrier gas. In the growth process, the tube furnace system followed the predefined temperature-rising program, in which the heating center was first set to ramp from RT to 380 °C with a heating rate of 20 °C min$^{-1}$ and was maintained at 380 °C for 30 min, and then continue to heat to 520 °C within 10 min and was stabilized for 3 min to fulfill the epitaxial growth process. To meet the requirement of optical characterization and transient absorption measurements, the supporting substrate ($SiO_2$/Si) is replaced with double-polished sapphire, other experimental conditions are kept unchanged.

### Synthesis of organic-inorganic hybrid halide perovskite/$WSe_2$ heterostructures

In the case of hybrid perovskite $MAPbI_3$ (where $MA^+$ is Methylammonium), $PbI_2$ powder was put in the center of the tube furnace (360 °C) while Methylamine hydroiodide (MA·HI) and substrates ($WSe_2$/$SiO_2$/Si) were located in the up-and down-stream zones, respectively. The growth process was carried at pressure of 200 mbar under the flow of argon gas of 60 sccm. The growth process was then terminated after 2 min growth at 360 °C and the system naturally cooled down to RT.

### Synthesis of all-inorganic halide perovskite/$SnS_2$/$WSe_2$ heterostructures

The multi-heterostructure synthesis followed the same procedure used for synthesis of all-inorganic halide perovskite/2D semiconductor heterostructures, only the substrate is replaced by $SnS_2$/$WSe_2$/$SiO_2$/Si substrate.

### Structural characterization

The morphologies of all samples were characterized by optical microscopy (Olympus TOKYO 163-0914, Japan), field emission scanning electron microscope (FE-SEM) (MIRA3 LMH, Hunan Navi New Materials Technology), and Atomic Force Microscope (AFM) (Bruker

Dimensional Icon system). To examine the cross-sectional morphology and chemical composition distribution of $CsPbI_3$/$WSe_2$ heterostructure, a focused ion beam (FIB) was introduced to prepare cross-section SEM specimens. The selected region of the heterostructure was protected by depositing and curing the organometallic Pt precursor to form a Pt-C protective layer on the perovskite surface and then was sectioned by using a few KV (2–5 KV) Ga ion beam milling to expose the cross-section profile. High-angle annular dark-field scanning transmission electron microscopy (HAADF-STEM) images were recorded on an aberration-corrected JEOL JEM-ARM200F with a cold-field emission gun at 200 kV. X-ray photoelectron spectroscopy (XPS, AXIS SUPRA system (KRATOS, Japan)) was conducted to characterize the chemical composition and band structure determination. In particular, ultraviolet photoemission spectroscopy (UPS) measurement was carried out in an ultrahigh vacuum chamber with a base pressure <10−11 Torr using a Phoibos-100 electron analyzer (Specs GMBH). For the latter measurement, a helium gas discharge lamp with photon energy of 21.2 eV served as a UV source and the samples were uniformly deposited on conductive substrates (highly oriented pyrolytic graphite and Au film). XRD (MiniFlex600 instrument with a Cu Kα radiation source ($\lambda = 0.15406$ nm)) was used to identify the crystal phase of epitaxial perovskite, operating at a voltage of 40 KV and a current of 200 mA. In detail, to accurately identify the phase of epitaxial square perovskite, large-area highly oriented pyrolytic graphite was chosen as a van der Waals substrate for the selective growth of square/rectangle perovskite to exclude the influence from 3D micropyramids.

## DFT Calculation

All the DFT calculations were performed in Vienna ab initio Simulation Package (VASP)[76]. The generalized gradient approximation parameterized by Perdew-Burke-Ernzerhof (GGA-PBE) was used to describe exchange-correlation functional[77]. Projector augmented wave (PAW) method was used to describe the electron-ion interactions and the cutoff energy for the plane wave basis set is 500 eV. The van der Waals interaction was treated by DFT-D3 method[78]. The convergence energy threshold for the electronic self-consistent iteration is $10^{-5}$ eV and the forces threshold is 0.03 eV Å$^{-1}$. $2 \times 1$ supercell of $CsPbI_3$ (001) surface and $3\sqrt{2} \times 3$ surpercell of $WSe_2$ (0001) surface was used to build $CsPbI_3$/$WSe_2$ heterostructures and the vacuum layer of 15 Å was introduced to avoid interactions between periodic images. The Brillouin zone integration sampling was using $1 \times 1 \times 1$ k-points meshes for the heterostructures model. The structure optimization convergence criterion for the electronic self-consistent field loop was set to $1 \times 10^{-5}$ eV atom$^{-1}$.

The adsorption energy ($E_{ads}$) is calculated following the equation:

$$E_{ads} = E_{slab + mol} - (E_{slab} + E_{mol})$$

where $E_{slab + mol}$ is the total energy of the surface with molecule adsorption, $E_{slab}$ and $E_{mol}$ represent the energy of the clean surface and the molecule, respectively. The $CsPbI_3$ diffusion pathways calculations were employed using the climbing image nudged elastic-band method.

## Optical measurements

The Raman and steady-state photoluminescence (PL) spectra were obtained by a confocal microscope-based Raman system (inVia Reflex, Renishaw, with a 488 nm laser as the excitation source). Time-resolved photoluminescence (TRPL) spectra were acquired using a fluorescence spectrophotometer equipped with time-correlated single-photon counting spectra under a 488 nm laser excitation. Steady-state absorption spectra was obtained by a micro-UV−vis absorption spectroscopy equipped with Metatest ScanPro Laser Scanning System (ScanPro Advance, Metatest). The femtosecond

transient absorption setup used for this study was performed using a commercial pump-probe setup (TA-100, Time-Tech Spectra, Co., Ltd.), which is based on a regenerative amplified Ti: sapphire laser system from Coherent (800 nm, 190 fs and 1 kHz repetition rate), nonlinear frequency mixing techniques and the Femto-TA100 spectrometer. Briefly, the 800 nm output pulse from the regenerative amplifier was split into two parts with a 50% beam splitter. The transmitter part was used to pump a TOPAS Optical Parametric Amplifier (OPA) which generates a wavelength-tunable laser pulse from 250 nm to 2.5 μm as pump beam. The reflected 800 nm beam was split again into two parts. One part with less than 10% was attenuated with a neutral density filter and focused into a 2 mm thick sapphire window to generate a white light continuum from 480 nm to 900 nm used for probe beam. The probe beam was focused with an Al parabolic reflector onto the sample. After preparing the sample, the probe beam was collimated and then focused into a fiber-coupled spectrometer with CMOS sensors and detected at a frequency of 1 kHz. The delay between the pump and probe pulses was controlled by a motorized delay stage. The pump pulses were chopped by a synchronized chopper at 500 Hz and the absorbance change was calculated with two adjacent probe pulses (pump-blocked and pump-unblocked). The transient reflection was conducted on the same pump-probe setup with reflection mode.

The lasing characterization was performed based on the confocal μ-PL system (WITec, alpha-300). A mode-locked Ti: Sapphire laser (Tsunami, 800 nm, pulse width 100 fs) was amplified by a regenerative amplifier laser (Spitfire Ace 100, 1 kHz) and then was introduced into an optical parameter amplifier (OPA, TOPAS Prime). The output laser from OPA can be continuously tuned from 300 nm to 2600 nm and used as the excitation source. The lasers at 400 nm (pulse width 100 fs, repetition frequency 1 kHz) were used for pumping the perovskite samples. Both lasers were introduced to the confocal system and focused onto the samples through a 50× objective lens with an expanded laser spot size to uniformly excite the whole nanoplate, minimize heat and optical damage at high energy pumping conditions, and ensure sufficient energy injection. The emission was collected with the same objective. The power at the input was altered by the neutral density filters. For collecting the lasing images, the emission signals were imaged on a CCD camera with a long-pass filter to block the laser line. The polarization ratio was obtained from the lasing spectra recorded at different rotation angles of a polarizer placed in front of the spectrometer. All the optical measurements were performed at room temperature under ambient conditions.

## Reporting Summary

Further information on research design is available in the Nature Portfolio Reporting Summary linked to this article.

# Data availability

Relevant data supporting the key findings of this study are available within the article and its Supplementary Information file. All raw data generated during the current study are available from the corresponding author upon request.

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

## Acknowledgements

We acknowledge the support from the National Key R&D Program of the Ministry of Science and Technology of China (grant no. 2022YFA1203801), the National Natural Science Foundation of China (grant nos. 51991340, 51991343, 52221001), the Hunan Key R&D Program Project (grant no. 2022GK2005).

## Author contributions

X.D. and L.Z. conceived and designed the research. L.Z., Y.W., M.L., Z.Z., X.S., B.L., C.Y., and Y.L. contributed to the synthesis of the 2D semi-conductors and halide perovskite/2D semiconductor heterostructures. L.Z., X.L., M.L., C.Y., and Z.Z. contributed to the Raman, PL, and HAADF-STEM characterizations. R.S. performed the AFM and SEM characterization. L.Z., A.C., and X.Z. helped with the optical measurements. L.Z. and X.D. co-wrote and revised the manuscript. All authors contributed to the discussion and commented on the manuscript.

## Competing interests

The authors declare no competing interests.
