## [Peer Review File · Nature Communications]

Facet-selective growth of halide perovskite/2D semiconductor van der Waals heterostructures for improved optical gain and lasingREVIEWER COMMENTS

Reviewer #1 (Remarks to the Author):

In this work, the authors demonstrated a method to growth halide perovskite/2D transition metal dichalcogenides heterostructures and developed corresponding optoelectronic applications in Lasing. The heterostructures were formed via van der Waals epitaxy, and detailed growth mechanisms were proposed. It is found the metal halide perovskites grow with a preferred orientation and lattice plane on the surface of 2D layer. The orientation action of epitaxial layers can be attributed to the thermodynamically favored exposure of plane facets with minimum energy. In general the work is systematically organized. However, regarding the growth and applications of 2d heterostructures there are still several issues should be fully address before publication.

(1) Temperature dependent growth behaviors need more detailed analysis and discussion. It was mentioned in the manuscript that the epitaxial nucleation density is decreased with temperature increasement. The authors are suggested to give more specific explanation on this observation. Commonly with increasing temperature the surface interaction between precursors and growth templates should be increased, and result in an improved nucleation density. Or in the growth process there is a competition between the strength of surface interaction and adatom diffusion energy barrier?

(2) Since the temperature range is around 510 °C to 520 °C. It is not a big temperature difference. In the growth furnace such temperature difference can be affected by many factors including gas flow rate, position of samples, heating rate and etc. Therefore, to reach a more solid conclusion, wider temperature range for the growth should be investigated.

(3) Although it is considered as a perfect atomically flat surface for the 2D WSe₂ layer, most of the 2D layers synthesized by CVD method may include surface defects (either transition metal or chalcogen vacancies). Such defects may serve as nucleation sites for metal halide growth. At higher temperature, these defects may also react with the precursors for metal halide growth (get the defects repaired by Br or I atoms). Therefore, beside considering the thermodynamical influence, it is also necessary to analyze the kinetic factors from the 2D template (such as the generation/healing of WSe₂ defects during epitaxial growth).

(4) Several previous reports have shown the advantages of formation 2D metal halide/ transition metal dichalcogenides heterostructures, especially for optoelectronic applications. However, the preparation method has not been well established. Therefore, the controllability for the growth of such 2D heterostructure is critical. However, it seems in this work the thickness and location of the epi-layer is lack of control. The authors are suggested to add discussions on how to reach a better control of the growth. Otherwise, new contribution on the understanding of 2D heterostructure growth is rather limited.

(5) The authors have analyzed orientation of epi-layer on 2d surface. It is also interesting to discuss whether the lateral size of epi-layer is limited by the underlying materials. And since the metal halide is a 3D crystal, what is the driving force or main influence for the in plane and out of plane growth that determining the lateral size and thickness of metal halide perovskite layer.

Although the lateral size distribution was provided in Figure S11 and the diffusion length on 2D surface was analyzed. One fundamental issue here is that the number of crystals and thickness on each 2D flakes were not well considered. It is also seen that there are more metal halides nucleations at the edge of 2D WSe₂ layers, which may compete the precursors for metal halide growth. Therefore, I am wondering whether the data regarding the lateral size distribution is reliable. Could the authors use 2D WSe₂ layer with much improved surface area for this study. So that metal halide perovskites grown at the edge of WSe₂ would less affect the growth of epi-layer and help to reveal the growth mechanism.

(6) Technical issues: several typo should be corrected, for example:
on page 6, “th XRD patterns, four dominat...”;
on page 7, “... can well match with three unit cells of WSe₂ ...”;

Reviewer #2 (Remarks to the Author):

This study presents a significant advancement in the synthesis of halide perovskite/2D semiconductor heterostructures using CVD van der Waals epitaxy for potential photonic applications. The research showcases the benefits of this approach, including high quality, robustness, and scalability. By investigating the epitaxial growth of halide perovskite and 2D semiconductor materials through a combination of experimental analysis and theoretical calculations, the study reveals a reduction in defect density and a more uniform energy landscape in the van der Waals epitaxial perovskite semiconductors. These improvements lead to enhanced optical gain properties, resulting into ultralow-threshold and stable single-mode lasers. The findings demonstrate the potential of mixed-dimensional heterostructures for on-chip light sources and integrated optoelectronics devices. However, several key revisions are recommended to address as following.

(1) The underlying 2D semiconductor in-plane edges, some irregular crystals appeared in the Figure 1 (b) and 1(d), the authors need provide a reasonable explanation (possible mechanism) for this growth behavior?

(2) In the robust and universal synthesis of halide perovskite/2D semiconductor heterostructures, what are the limiting factors for the reliable synthesis of heterostructures?

(3) Can the heterostructures, such as CsPbI₂Br/WSe₂ demonstrated for lasing application, directly synthesized on Si substrate for silicon photonic integration?

(4) There is an error in the labeling of the y-axis of Figure 5e,f,g.

(5) The writing and English should be improved for better reading and understanding.

Reviewer #3 (Remarks to the Author):

Liqiang Zhang et al. demonstrated a methodology for synthesizing heteroepitaxy of halide perovskite on 2D TMDCs. From the heterostructures, they observed an improved gain coefficient, reduced gain threshold and prolonged gain lifetime. Under optical pumping, the halide perovskites exhibit "laser-like" emission. The authors may consider addressing the following comments:

1) Figure 1f, 1g and 1h indicate that the heterostructure has lower emission efficiency compared with individual halide perovskites. This suggests that heterostructures may not be suitable for high performance luminescence applications. Could the authors provide motivations for using them for lasing applications?

2) The claim of "Threshold is much lower" may not be accurate when comparing 2.21 with 6.36 $\mu\text{J}/\text{cm}^2$. The authors may need to reconsider their claims throughout the manuscript, such as "highly reproducible...", "largely reduced...", "excellent" and so on.

3) Light is a wave with both amplitude and phase. Laser light is coherent. To claim a sharp peak to be lasing, only showing data on amplitude changing with pump power is insufficient. Could the authors provide lasing data on phase, such as the temporal coherence (or spatial coherence), or quantum photon statistics data?

Reviewer #1 comments and author response

General Comment: In this work, the authors demonstrated a method to growth halide perovskite/2D transition metal dichalcogenides heterostructures and developed corresponding optoelectronic applications in Lasing. The heterostructures were formed via van der Waals epitaxy, and detailed growth mechanisms were proposed. It is found the metal halide perovskites grow with a preferred orientation and lattice plane on the surface of 2D layer. The orientation action of epitaxial layers can be attributed to the thermodynamically favorited exposure of plane facets with minimum energy. In general the work is systematically organized. However, regarding the growth and applications of 2d heterostructures there are still several issues should be fully address before publication.

Authors Response: We sincerely thank the reviewer for the very careful review and valuable suggestions on van der Waals epitaxial growth dynamics and mechanism. Overall, these insightful and constructive comments have motivated us to conduct additional studies and analyses, and helped to significantly strengthen the quality of our manuscript. All the comments have been addressed accordingly and the details of revision can be seen below.

Specific comments:

Comment #1-1: Temperature dependent growth behaviors need more detailed analysis and discussion. It was mentioned in the manuscript that the epitaxial nucleation density is decreased with temperature increasement. The authors are suggested to give more specific explanation on this observation. Commonly with increasing temperature the surface interaction between precursors and growth templates should be increased, and result in an improved nucleation density. Or in the growth process there is a competition between the strength of surface interaction and adatom diffusion energy barrier?

Authors response: We sincerely thank the reviewer for the constructive comments and suggestions. However, we would like to kindly draw your attention to the content in the Figure S12-13, page of 14-21of the Supplementary Information. In this part, the nucleation density, side length, growth rate and crystal morphology transformation were systematically investigated at epitaxy-accessible temperature window through the combination of experimental and theoretical approaches. We sincerely apologize that our manuscript might have been unclear in stating the relationship between epitaxial nucleation/growth and temperature. In order to clarify this statement, we provided detailed discussions and key insights to emphasize the topic and enhance the clarity. In response to the constructive feedback, we added detailed discussion about temperature-dependent nucleation and growth behaviors in the section of “Mechanism of oriented epitaxial growth of CsPbI₃ on monolayer WSe₂” at the page 9-10 of the main text in the revised manuscript.

Page 9-10 in the main text

The unique energy landscape of van der Waals surface essentially modifies the epitaxial nucleation and growth process. More specifically, when increasing the

temperature, the thermal energy-driven diffusion length is greatly prolonged, the long-range migration of adatoms to energetic favorable sites for large critical-size nucleation formation is kinetically favored. Simultaneously, the higher kinetic energy will inevitably give rise to much easier desorption of adatoms. Consequently, simultaneous collisions of arriving adatoms on the inert surface are greatly suppressed, which leads to large nuclei with reduced density. By contrast, at low temperature, the diffusion length with lower kinetic energy becomes shorter and the corresponding desorption probability of adatom is reduced. This scenario thus offers a high probability of collision among diffusing adatoms, thus leading to smaller nuclei with increased quantity. The subsequent in-plane growth shares the same driving force as nucleation via van der Waals forces, which proceeds through diffusion and registry of the adatoms to the growing front of stable nuclei. At high temperature, in addition to the long mean-free-path of adatom diffusion, the high local supersaturation gradient is synchronously introduced. Once the stable nuclei formed, adatoms can easily and quick transport to the edges for continuously feeding the in-plane epitaxial growth. Due to the sparser nuclei distribution, the single nuclei can receive sufficient nutrient atoms, thus accelerating the growth rate. In the case of lower-temperature growth, the opposite case happens.

In the growth process, a competition between the strength of surface interaction and adatom diffusion energy barrier is not the dominant factor in governing the epitaxial growth. The main reasons may originate from the following facts.

Firstly, the dangling bonding-free inert surface of 2D semiconductor leads to weak interface adhesion between the adatoms and 2D semiconductor surface. The strength is typically characterized by the weak van der Waals energy. Consequently, the van der Waals surface provides extremely low diffusion barrier to adatom, which suggests that the transport and addition of adatom to the growing front at the epilayer for feeding growth is not limited by kinetic barriers during epitaxy.

Secondly, the fractal morphology in epitaxial perovskite materials is a result of the competition between van der Waals diffusivity and thermodynamic driving force (Cryst Growth Des. 15, 4741–4749, (2015)), which is a typical indicator of a competition between the strength of surface interaction and adatom diffusion energy barrier. However, no fractal morphology of resultant crystal can be observed in our van der Waals epitaxy scheme, signifying the absence of such competition in the growth process.

Thirdly, in fact, the van der Waals epitaxial growth is fundamentally dictated by edge growth through the competition between adsorption and desorption of precursor molecules at growth fronts. (Adv. Mater 33, 2105079 (2021)). Therefore, the growth temperature should be carefully optimized for controlling the in-plane growth of epilayer. The reason is that the rate of absorption and deabsorption should be balanced owing to the low adsorption energy.

Comment #1-2: Since the temperature range is around 510 °C to 520 °C. It is not a big temperature difference. In the growth furnace such temperature difference can be affected by many factors including gas flow rate, position of samples, heating rate and

etc. Therefore, to reach a more solid conclusion, wider temperature range for the growth should be investigated.

Authors Response: We sincerely thank the reviewer for the constructive comments and suggestions. Indeed, we agree that wider temperature range for the growth is better for investigating the growth dynamics. We actually try our best to expand the temperature range to observe the growth dynamics, but we find that when temperature lower than 510 °C or higher than 520 °C, the perovskite is unable to successfully grow. Specifically, when the growth temperature is set to be lower than 510 °C, no particles or film are deposited on WSe₂ monolayer surface. This is because such low temperature is unable to afford a sufficient flux of precursor in the vapor phase over a distance between the precursor sources to the growth substrate. When the growth temperature is higher than 520 °C, the chaotic films partially cover the monolayer WSe₂ surface. The films are non-emissive, indicating no perovskite formation. The microscopic mechanism lies in the formation process of halide perovskite in physical vapor depositions. The nucleation formation and growth need the intercalation of Cs⁺ cations into metal halide clusters ([PbI₆]⁴⁻ octahedron) (APL Mater. 9, 100703 (2021)). The lower temperature may not support required thermodynamic driving force for intercalation, while higher temperature induces fast adatoms (Cs⁺ and [PbI₆]⁴⁻) migration due to low diffusion barrier energy, which reduces the simultaneous collision events, thus limiting reaction probability. These effects may not afford stable nucleation formation and sequent growth.

In the growth furnace, carrier gas flow rate is collectively and precisely controlled by gas flow indicator and needle valve to keep steady flow, position of sample (substrate) is fixed and heating rate is optimized for minimized fluctuation. So, the temperature-dependent growth observations have good repeatability for probing the growth dynamics and mechanism.

Comment #1-3: Although it is considered as a perfect atomically flat surface for the 2D WSe₂ layer, most of the 2D layers synthesized by CVD method may include surface defects (either transition metal or chalcogen vacancies). Such defects may serve as nucleation sites for metal halide growth. At higher temperature, these defects may also react with the precursors for metal halide growth (get the defects repaired by Br or I atoms). Therefore, besides considering the thermodynamical influence, it is also necessary to analyze the kinetic factors from the 2D template (such as the generation/healing of WSe₂ defects during epitaxial growth).

Authors response: We sincerely thank the reviewer for the constructive comments and insightful suggestions. Following the reviewer's suggestions, we made qualitative and quantitative analysis and explanation on generation/healing of WSe₂ defects during epitaxial growth. In the revised manuscript, we added the corresponding discussion in Figure S4, Page 6 in the Supplementary Information.

Figure S4, Page 6 in the Supplementary Information

The generation/healing of WSe₂ defects during epitaxial growth is most unlikely

to occur. The qualitative and quantitative analysis and explanation are shown below. Qualitatively speaking, a thorough literature survey demonstrates that the substantial defects, strain and doping effect introduced from halide interaction mainly take place at temperature range of more than 700 °C (Nature Materials 17, 535–542 (2018); Applied Materials Today 1, 60–66 (2015); Sci. Adv. 7. eabj3274 (2021)). Our growth temperature (500 °C) is far lower than that the reported to be more than 700 °C, thus ruling out the possibility of generation of defects within monolayer WSe₂ in our epitaxial growth process. Additionally, the halide species could preferentially passivate the edges of the 2D WSe₂ monolayer due to their prominent dangling bond states. This edge passivation effect may relax in-plane strains to suppress defect generation. (ACS Nano 14, 6570–6581 (2020)).

To quantitatively validate the assumption, the Raman, Raman mapping, photoluminescence, photoluminescence mapping and time-resolved photoluminescence experiments were performed (Sci. Adv. 5, eaau4728 (2019); Nano Res. 14, 29–39(2021)). The Raman mapping (Figure R2 (d) and (e)) and photoluminescence mapping (Figure R2 (g) and (h)) results show a very uniform intensity distribution, suggesting the absence of defect generation (If defects are generated during epitaxy growth, the mapping image is non-uniformity since random distribution nature of defect). Furthermore, The Raman and PL spectra (Figure R2 (f) and (i)) are collected from monolayer WSe₂ before perovskite epitaxy, after epitaxy and hetero-interfacing area after washing perovskite. The peak position and intensity are almost identical, signifying no defect generation/healing of WSe₂ monolayer during epitaxial growth (ACS Nano 14, 6570–6581 (2020)). Besides, due to the sensitivity of photoluminescence to defect, the time-resolved photoluminescence results (Table R1) are employed to qualify this process. The lifetimes of monolayer WSe₂ before epitaxial growth and after epitaxial growth experience almost unaltered, robustly indicating good crystal quality retention of monolayer WSe₂ during epitaxy growth process (Sci. Adv. 5, eaau4728 (2019)). These spectroscopic characterization and discussion were provided in Figure S4, Page 6 in the Supplementary Information.

Figure R2 (Figure S4 in the Supplementary Information). Monolayer nature of as-synthesized WSe₂ and its stability during CsPbI₃ epitaxy. Atomic force microscope (AFM) image and phase image of as-synthesized WSe₂ on SiO₂/Si substrate (a, b). Time-resolved photoluminescence (TRPL) experimental data and fitting results of WSe₂ before and after CsPbI₃ epitaxy (c). Raman mapping/spectra (e, f) and PL mapping/spectra (h, i) of WSe₂ at different conditions: before and after CsPbI₃ epitaxy and contacting region of CsPbI₃ after washing away CsPbI₃ using acetone. Scale bar $\approx 15 \mu\text{m}$. In Figure c, f and i, the before and after are the abbreviations for “before CsPbI₃ epitaxy” and “after CsPbI₃ epitaxy”, respectively.

Table R1 (Table S1 in the Supplementary Information) Detailed fitting parameters of TRPL results of monolayer WSe₂ before and after CsPbI₃ epitaxial growth

Samples	A ₁ (%)	τ_1 (ns)	A ₂ (%)	τ_2 (ns)
WSe ₂ (before)	0.90	0.041	0.10	0.34
WSe ₂ (after)	0.87	0.036	0.13	0.33

Comment #1-4: Several previous reports have shown the advantages of formation 2D metal halide/transition metal dichalcogenides heterostructures, especially for optoelectronic applications. However, the preparation method has not been well established. Therefore, the controllability for the growth of such 2D heterostructure is critical. However, it seems in this work the thickness and location of the epi-layer is lack of control. The authors are suggested to add discussions on how to reach a better control of the growth. Otherwise, new contribution on the understanding of 2D heterostructure growth is rather limited.

Authors response: We sincerely appreciate the reviewer's insightful suggestion about adding a discussion on how to reach a better control of the thickness and location of epilayer. In the revised manuscript, we added the corresponding discussion in Figure S12-13, Page 21 in the Supplementary Information.

At present work, we have revealed the growth mechanism regarding facet/alignment-specific growth, and demonstrated the growth controllability in nucleation density and lateral length by tuning critical growth parameters. The thickness control strategy mainly involves two ways: growth temperature and growth time. Firstly, due to the nonlayered crystal structure and ionic-bonding soft-lattice nature in 3D perovskite, the growth along vertical direction is more thermodynamically controlled. At low temperature, the perovskite thickness is relatively thin. At high temperature, the thickness tends to become thick. The detailed investigation about the relationship between thickness and growth time is presented in Figure R3 (k). Secondly, in general, at a given growth temperature, the longer the growth time is, the thicker the perovskite is. The location of the epi-layer is naturally governed by defect-induced nucleation site of underlying monolayer semiconductor. The specific alignment is intrinsically controlled by energetically favorable degenerate states at three-fold symmetry of underlying monolayer semiconductors. These epitaxial growth actions are direct indicators of intrinsic epitaxial interaction between halide perovskite and 2D semiconductor, and also are our primary research focus for establishing a guiding reference for designing halide perovskite/2D semiconductor heterostructures.

Figure S12-13, page 21 in the Supplementary Information

To offer more contribution on the understanding of such hybrid heterostructure growth, we try our best to provide some suggestions on the growth controllability. In term of thickness, it should be tailored based on the specific optoelectronic applications. For light-harvesting optoelectronic (photovoltaic and photo-detection), the thickness should be controlled from 100 nm to 500 nm in order to maximize light absorption for improving efficiency and sensitivity. This can be simply controlled by growth temperature and time. For optical-addressable excitonic and quantum devices, the thickness should be controlled as thin as possible because ultrafast energy transfer, strong interfacial coupling and manageable exciton dynamics are much more favorable. This can be manipulated by space-confined growth method and simultaneous reduction of growth temperature and time. As for the location control of epi-layer, the strategies may include laser-defined defect-induced nucleation, hetero-nucleation-defining

approach and pre-defining 2D material patterns. The location controllability is critical to fabricate heterostructure arrays, which is an important goal for practical optoelectronic applications.

Comment #1-5: The authors have analyzed orientation of epi-layer on 2d surface. It is also interesting to discuss whether the lateral size of epi-layer is limited by the underlying materials. And since the metal halide is a 3D crystal, what is the driving force or main influence for the in plane and out of plane epilayer growth that determining the lateral size and thickness of metal halide perovskite layer.

Although the lateral size distribution was provided in Figure S11 and the diffusion length on 2D surface was analyzed. One fundamental issue here is that the number of crystals and thickness on each 2D flakes were not well considered. It is also seen that there are more metal halides nucleations at the edge if 2D WSe₂ layers, which may compete the precursors for metal halide growth. Therefore, I am wondering whether the data regarding the lateral size distribution is reliable. Could the authors use 2D WSe₂ layer with much improved surface area for this study. So that metal halide perovskites grown at the edge of WSe₂ would less affect the growth of epi-layer and help to reveal the growth mechanism.

Authors response: We sincerely thank the reviewer for the constructive comments and valuable suggestions. We address your insightful comments by carefully analyzing related growth thermodynamic and kinetic process as well as extensive experimental measurements. Specifically, we firstly provide in-depth insights into whether the lateral size of epi-layer is limited by the underlying materials from the perspective of growth kinetics. Following your insightful suggestions, we then carried out corresponding experiments using large-area monolayer WSe₂ as growth substrate. The corresponding determining factors that govern the in-plane and out-of-plane growth are deeply discussed. The point-by-point answers are as follow. In the revised manuscript, we have added the important information in Figure S12-13, Page 17-18 of the Supplementary Information.

Figure S12-13, page 17-18 of the Supplementary Information

Whether the lateral size of epi-layer is limited by the underlying 2D materials mainly depends on the nucleation location. If the initial nucleation site is close to the edges of monolayer WSe₂, the lateral growth rate is suppressed. If the initial nucleation sites keep their certain distance from the edge of monolayer WSe₂, the lateral growth is impervious to the presence of the edge. In the former case, the dangling bond sites at the edge of the monolayer WSe₂ act as capturing sites to influence the mass transport and change the distribution of perovskite precursor ions in the growth process. The competitive depletion of precursor ions at edge thus suppresses the nourishment concentration for continuous growth. The characteristic distance is determined by diffusion-controlled length scale (l_d) based on the DDA (deposition, diffusion, and aggregation) model. In the growth model, $l_d \approx (D/F)^{1/6}$, where l_d is the distance of the region where no deposition occurs, D is a diffusion constant, and F is a flux of monomer (Sci. Adv.4. eaat2390 (2018)). At the edge of monolayer WSe₂ and perovskite

nuclei at the edge, the adatoms are intrinsically attached to the active dangling bond states. D is thus very small and the length scale of depletion is rather limited to their neighboring region. Therefore, the nuclei far from the edge are free from the influence the edge of underlying materials.

To minimize the effect of edge states of monolayer semiconductor, we expand the surface area of 2D WSe₂ for growing the halide perovskites. The surface area of WSe₂ monolayer is as large as 7800 μm² and 2 mm² for single crystal domain and polycrystalline film, respectively. The experimental results are shown in Figure S12 (f) and (g). Although polycrystalline nature of large-area WSe₂ film, the halide perovskites still kept the facet and orientation growth selectivity, suggesting that the grain boundary has insignificant effect on lateral size of perovskite growth. We then make a statistical investigation on the lateral size distribution as a function of growth time. The number of investigated perovskite nanoplates is more than 600, which ensure enough sample group for obtaining the reliable results. The results presented in Figure S12 (h), (i) and (j) clearly show that the lateral length distribution follows the similar trends with ordinary size WSe₂ monolayer in Figure S12 (c), (d) and (e). Specifically, the average lateral length of the perovskite domain is estimated to be 12.2 ± 1.85 μm, 22.4 ± 2.1 μm and 28.4 ± 2.15 μm within 2, 3 and 4 minutes. The lateral length within longer growth time (3 min and 4 min) is a bit larger than those observed in Figure S12 (d) and (e). This optimized investigation demonstrated that halide perovskite grown at the edge of monolayer WSe₂ has a slight effect on the in-plane perovskite growth by competitive depletion of feeding precursor.

The in-plane epilayer growth on 2D semiconductor is a kinetically controlled process, where the growth temperature is the main influencing factor. In general, 2D substrates provide low diffusion barriers and adhesion energy to adatoms due to the dangling bond-free inert surface. Therefore, adatoms can quickly diffuse and add on to the fastest growth front of the growing epilayer to extend it. At high temperature, adatoms with high kinetic energy have the increased diffusion rate. Consequently, adatoms can easily and faster migrate to the growth edge for continuously feeding the in-plane epitaxial growth of epilayer. Additionally, the high local supersaturation gradient is synchronously introduced. The single nucleation can receive sufficient nutrient atoms, thus accelerating the growth rate. When adatoms deposited on 2D material surface with insufficient thermal energy at lower temperature, limited adatoms can arrive at the growing front of nucleus, thus reducing the in-plane growth rate. It is should be noted that the growth temperature should be optimized for controlling the in-plane growth of epilayer. The reason is that the rate of absorption and deabsorption should be balanced owing to the much lower adsorption energy.

The out-of-plane growth of 3D perovskite is intrinsically driven by the non-van der Waals bonding nature of the 3D nonlayered perovskite. The ionic-bonding lattice networks lead to low cohesive energy of perovskites, which makes the out-of-plane growth more sensitive to growth temperature. The out-of-plane growth are composed

Figure R3 (Figure S12 in the Supplementary Information) Experimental observations of epitaxial nucleation and growth evolution. (a) Epitaxial nucleation density as a function of growth temperature. (b) Crystal growth rate obtained by the edge length as a function of growth time relationship. The statistical distribution of the lateral width of CsPbI₃ plates obtained with growth times of 2 min (c), 3min (d) and 4min (e). To minimize the edge nucleation competition effect of WSe₂ monolayer, the large-area single crystal WSe₂ (edge length of about 300 μm in Figure (f), scale bar: 50 μm) and polycrystalline monolayer WSe₂ (edge length of as large as 700 μm in Figure (g), scale bar: 100 μm) are selected as growth templates for observing the lateral size distribution as a function of growth time. The statistical distribution of the lateral length of perovskite nanoplates obtained with growth times of 2 min (h), 3min (i) and 4min (j) on large-area WSe₂ monolayer. (k) Temperature-dependent thickness evolution.

of two stages: new round of nucleation on the first perovskite layer and consequent growth along vertical direction. The vertical growth process is much more thermodynamically favorable. The temperature-dependent thickness evolution is investigated and shown in Figure S12 (k).

Comment #1-6: Technical issues: several typo should be corrected, for example:

on page 6, “th XRD patterns, four dominat...”;

on page 7, “... can well match with three unit cells of WSe₂ ...”;

Authors response: We sincerely thanks for the reviewer for the very careful review. We have carefully scrutinized the manuscript, and made corresponding revisions including some typos, chemical formula typo and long sentences. This manuscript has been well revised according to the reviewers' constructive suggestions.

Reviewer #2 comments and author response

General comment:

This study presents a significant advancement in the synthesis of halide perovskite/2D semiconductor heterostructures using CVD van der Waals epitaxy for potential photonic applications. The research showcases the benefits of this approach, including high quality, robustness, and scalability. By investigating the epitaxial growth of halide perovskite and 2D semiconductor materials through a combination of experimental analysis and theoretical calculations, the study reveals a reduction in defect density and a more uniform energy landscape in the van der Waals epitaxial perovskite semiconductors. These improvements lead to enhanced optical gain properties, resulting into ultralow-threshold and stable single-mode lasers. The findings demonstrate the potential of mixed-dimensional heterostructures for on-chip light sources and integrated optoelectronics devices. However, several key revisions are recommended to address as following.

Authors response: We sincerely thank the reviewer for the very careful review and valuable suggestions. We particularly appreciate the comments on the general applicability of your approach to photonics-compatible substrates. Based on the reviewer’s comments, we have conducted corresponding experiments on growing

CsPbI₂Br/WSe₂ heterostructures on photonic-compatible substrates, including Si and LiNbO₃, and on transparent substrate, double-polished sapphire. In the revision, we carefully addressed all the comments, as discussed below.

Specific comments:

Comment#2-1: The underlying 2D semiconductor in-plane edges, some irregular crystals appeared in the Figure 1 (b) and 1(d), the authors need provide a reasonable explanation (possible mechanism) for this growth behavior?

Authors response: We sincerely thank the reviewer for the constructive comments and valuable suggestions. We incorporated the explanation about irregular crystal growth behavior at the in-plane edge of monolayer WSe₂ in Figure S12-13, Page 15 in the Supplementary Information.

Figure S12-13, page 15 in the Supplementary Information

The edge-induced nucleation and growth behavior is common in 2D material-based heterostructure growth. The underlying growth mechanism lies in the nucleation and growth from the energy-favorable edge site of the monolayer WSe₂ (ACS Nano. 13, 885–893 (2019)). Specifically, the edge of monolayer WSe₂ is characterized by Se/W-terminated dangling-bond states. These dangling bonds act as strong absorption sites for capturing precursor radicals (Cs⁺ and [PbI₆]⁴⁻). The adatoms can randomly nucleate and then grow at edge. However, the disparate dangling-bond densities of the limited edge of monolayer WSe₂ and SiO₂ surface leads to imbalanced mass transport surrounding the adjacent regions and non-uniform adatom deposition at the heterointerface between SiO₂ and WSe₂ edge. The large fluctuation and frequently local disequilibrium of mass transport at WSe₂ edge finally result in an intertwined random nucleation and anisotropic growth scenario. The outcome is the irregular crystals located on WSe₂ edge (Materials Chemistry and Physics. 49, 93-104 (1997)).

Comment#2-2: In the robust and universal synthesis of halide perovskite/2D semiconductor heterostructures, what are the limiting factors for the reliable synthesis of heterostructures?

Authors response: We sincerely thank the reviewer for the constructive comments. In order to realize robust and universal synthesis of halide perovskite/2D semiconductor heterostructures, the limiting factors mainly consist of clean surface of monolayer semiconductor and temperature optimization. (1) The cleanness of 2D material surface has a profound influence on perovskite nucleation. The non-ideal surface will lead to the misorientation of nucleation and irregular nucleation. These eventually give rise to inferior crystallization with rich energetic disorders and uncontrolled facet-specific growth. (2) Growth temperature optimization can fundamentally impact both nucleation density and growth behavior. Specifically, in our extensive growth parameter optimization experiments, we found that the perovskite epitaxial growth temperature window is much narrower than reports on 2D material growth, because of extreme sensitivity of soft lattices of halide perovskite to temperature. The too lower and/or higher temperature all are unable to provide suitable environment for perovskite epitaxy.

The small temperature variation can lead to changes in nucleation density and growth rate.

Comment#2-3: Can the heterostructures, such as CsPbI₂Br/WSe₂ demonstrated for lasing application, directly synthesized on Si substrate for silicon photonic integration?

Authors response: We sincerely thank the reviewer for the constructive comments and suggestions. We highly agree with the reviewer (and the editor) that it is desirable to demonstrate the general applicability of halide perovskite/2D semiconductor heterostructures growth to different substrates. In response to the constructive feedback, we have carried out a set of experiments in epitaxially growing halide perovskite/2D semiconductor heterostructures on a broad range of substrates, including silicon (Si), lithium niobate (LiNbO₃) and transparent double-polished sapphire (Al₂O₃). The substrate selection criterions are based on the ready-to-integration photonic applications and optical characterization requirements. Si and LiNbO₃ are popular substrates for photonic integrated circuits, transparent sapphire is an ideal substrate for characterizing optical property.

In the revised manuscript, we have added the corresponding experimental results and discussion in the part of “Epitaxial growth of halide perovskite/2D semiconductor heterostructures” Page3, Page 4, Page 19 in the main text and Figure S2, Page 3-4 in the Supplementary Information.

Page 3 of the main text

Furthermore, the van der Waals epitaxial method has the general applicability to both CMOS-compatible substrate (SiO₂/Si) and photonic-compatible platforms (Si and LiNbO₃).

Page 19 of the main text

The epitaxial heterostructures can monolithically integrate on both CMOS-compatible substrate (SiO₂/Si) and ready-to-use photonic platforms (Si and LiNbO₃).

Page 4 of the main text

In addition to the growth compatibility with the silicon-based CMOS-compatible substrate (SiO₂/Si), the van der Waals epitaxy method has the general applicability to photonic-compatible platforms, such as Si and LiNbO₃, shown in Figure S2. The heterostructures are capable of robust and scalable growth on the ready-to-use photonic substrates, where the epitaxial perovskites still followed the facet/alignment-specific growth habit. This universal substrate-compatible monolithic integration of halide perovskite/2D semiconductor heterostructures shows great promises for on-chip photonic device applications.

Figure S2, page 3-4 of the Supplementary Information

The monolithically integrated halide perovskite/2D semiconductor heterostructures on selected substrates are shown below. The growth methodology and parameters are

identical to van der Waals epitaxy used for SiO₂/Si substrate. Figure S4 (a) displays the monolayer WSe₂ grown on Si substrate. Due to the subtle color contrast between monolayer WSe₂ and Si substrate surface, we adjust the color contrast of optical microscopy to visualize the monolayer WSe₂, which is marked by red dotted lines. The deposited WSe₂ on Si substrate is typical triangular single crystal domain. The resultant CsPbI₂Br/WSe₂ heterostructures are shown in Figure S2 (b), where rectangle halide perovskites are only epitaxially grown on monolayer WSe₂ surface and are well aligned with the edges of monolayer WSe₂. The pure Si surface is filled with irregular polycrystalline films and/or deposited precursor particles. This selective and oriented growth is consistent with the growth habit observed on SiO₂/Si substrate. Figure S2 (c) and (d) exhibit large-area monolayer MoS₂ film on LiNbO₃ (z cut) and their CsPbI₂Br/MoS₂ heterostructures. It is obvious that the rectangle perovskite single crystals are preferentially grown on monolayer MoS₂ surface, while pyramid-shaped perovskites only nucleated and grew on pure LiNbO₃ surface. This also strongly prove the robustness and scalability of van der Waals epitaxial integration. Figure S2 (e) is optical images of monolayer WSe₂ on double-polished sapphire substrate, the inset is a zoom-in optical image of single monolayer WSe₂ domain, which is manifested as a triangular single crystal domain. Figure S2 (f) shows CsPbI₂Br/MoS₂ heterostructures grown by van der Waals epitaxy. The epitaxial perovskite single crystals all are selectively grown and aligned on monolayer WSe₂ surface. The successful epitaxial growth of halide perovskite/2D semiconductor heterostructures on Si, LiNbO₃ and sapphire (Al₂O₃) substrates robustly demonstrate the general applicability to different substrates.

Figure R4 (Figure S2 in the Supplementary Information). The general applicability of our direct van der Waals epitaxy of halide perovskite/2D semiconductor heterostructures to different substrates. (a) Monolayer WSe₂ (triangle shape highlighted by red dotted lines) on Si substrate and (b) CsPbI₂Br/WSe₂ heterostructures grown on Si substrate using direct van der Waals epitaxy, scale bar: 50 μm. (c) Monolayer MoS₂ film on LiNbO₃ (z cut) substrate and (d) CsPbI₂Br/MoS₂ heterostructures grown on LiNbO₃ substrate using direct van der Waals epitaxy. Scale bar in (c): 100 μm and in (d): 50 μm. (e) Monolayer WSe₂ on double-polished sapphire substrate, inset is the zoom-in optical image of single monolayer WSe₂ domain, and (f) CsPbI₂Br/WSe₂ heterostructures grown on sapphire substrate. Scale bar: 50 μm.

Comment#2-4: There is an error in the labeling of the y-xylem of Figure 5e,f,g.

Authors response: We sincerely thank the reviewer for their detailed and thoughtful comments. We have corrected it.

Comment#2-5: The writing and English should be improved for better reading and understanding.

Authors response: We sincerely thank the reviewer for the very careful review. We revised the whole manuscript carefully to avoid language errors.

Reviewer #3 comments and author response

General comments: Liqiang Zhang et al. demonstrated a methodology for synthesizing heteroepitaxy of halide perovskite on 2D TMDCs. From the heterostructures, they observed an improved gain coefficient, reduced gain threshold and prolonged gain lifetime. Under optical pumping, the halide perovskites exhibit "laser-like" emission. The authors may consider addressing the following comments:

Authors response: We sincerely thank the reviewer for the constructive comments and suggestions. The reviewer's suggestions and criticisms help us substantially improve the quality of the manuscript. We have addressed the comments point-by-point as follows.

Specific comments:

Comment#3-1: Figure 1f, 1g and 1h indicate that the heterostructure has lower emission efficiency compared with individual halide perovskites. This suggests that heterostructures may not be suitable for high performance luminescence applications. Could the authors provide motivations for using them for lasing applications?

Authors response: We sincerely thank the reviewer for the insightful comments and suggestions. We are very sorry that we have not clearly stated our motivations for using heterostructures for lasing applications in our previous version. Our primary motivation and feasibility of the halide perovskite/2D semiconductor heterostructures for lasing application mainly come from the unique advantages of our proposed van der Waals heteroepitaxy integration scheme.

(1) Suitable photonic configuration and enhanced optical gain property. The epitaxial monocrystalline halide perovskites feature a well-defined photonic micro-cavity geometry with whispering gallery mode. This self-organized single-crystalline halide perovskite micro-cavity can simultaneously function as lasing gain media and optical feedback supplier to support optical amplification through stimulated emission with optical coherence properties within themselves, which eliminates the stringent requirements for micro-cavity post-processing with aggressive micro/nano-fabrication technologies. Furthermore, compared with the individual perovskite crystal, the epitaxial halide perovskite/2D semiconductor heterostructures are characterized by minimized energetic disorder (Urbach energy: 6.5 meV) and enhanced optical gain property, including improved gain coefficient (12000 cm^{-1}), reduced gain threshold ($6.0 \times 10^{17} \text{ cm}^{-3}$) and prolonged gain lifetime (98 ps). These characteristics are higher than that of individual perovskite and

previous reports, demonstrating that the epitaxial heterostructures are promising building blocks for high performance lasing application.

- (2) The tunable 2D semiconductor layer and clean interface are promising components for developing future perovskite-based electronically driven laser. Specifically, towards electronically driven laser, the basic configuration is composed of active optical gain medium and electron (hole) transporting layers. The semiconducting monolayer in our hetero-integrated heterostructures can be used as transporting layers for charge carrier injection because of their high carrier mobility, tunable polarity and ambient stability. Additionally, the epitaxial halide perovskite/2D semiconductor heterostructures have naturally-formed clean and ordered interface, which is beneficial for efficient charge carrier injection in laser diode. Therefore, demonstrating the lasing action in the hetero-heterostructures provide fundamental and guiding insights into the feasibility of such photonic building blocks for further laser design.
- (3) Front-end monolithic hetero-integration compatibility with ready-to-use substrates. Perovskite-based optoelectronics and photonics devices have long been suffered from hetero-integration compatibility issue due to huge lattice mismatch and strain effect from the disparate thermal expansion coefficient between substrate and perovskite. The van der Waals epitaxial halide perovskite/2D semiconductor heterostructures are able to monolithically integrate on both CMOS-compatible substrate (SiO_2/Si) and photonic-compatible platforms (Si and LiNbO_3), which pave the pathways for on-chip optoelectronics. In addition, the stability of lasing media is also enhanced, showing the unique practical advantage of van der Waals epitaxy integration. More broadly, using the same material for electronics and photonics in a single device/circuit could increase performances and reduce the power consumption of the integrated chip due to the electronically active feature of the heterostructures. Along with the front-end hetero-integration compatibility, this means that such functional material platform can be used to design photonic components and electronic units on single on-chip for reconfigurable photonic-electronic integrated devices, while the potentials for photonic lasing application have remained unexplored.

The mechanism responsible for the lasing action in our heterostructures is closely related with the ultrafast lasing dynamics, which is distinctly different from PL quenching. Specifically, the PL quenching in heterostructures originates from the long-range carrier diffusion and interfacial charge carrier transfer, where the diffusion-limited quenching process typically occurs within the time scale of nanosecond (ns). However, the lasing dynamics investigation (Figure 4 in the main text) shows that the onset of stimulated emissions take place at the time scale of picosecond (ps), which is several orders of magnitude faster than that of quenching effect. Therefore, the ultrafast lasing process is impervious to the presence of the 2D semiconductor layer, which acts as a charge carrier quencher (transport layer) (Nature Materials, 13, 476–480 (2014)). These in-depth insights has presented in the section of “Improved energetic disorder landscape in $\text{CsPbI}_2\text{Br}/\text{WSe}_2$ heterostructures”, page 11 in the main text.

In the revised manuscript, we have added the corresponding key motivations in the section of “Reduced laser threshold of CsPbI₂Br/WSe₂ heterostructure”, page 14-15 in the main text.

Page 14-15 in the main text

The above-mentioned investigations have shown that our epitaxial heterostructures have combined unique properties for photonic lasing application. Firstly, the epitaxial monocrystalline halide perovskites naturally yield a well-defined photonic micro-cavity geometry with whispering gallery mode, which can simultaneously function as lasing gain media and optical feedback supplier to support optical amplification through stimulated emission within themselves. More importantly, the epitaxial halide perovskite/2D semiconductor heterostructures are characterized by minimized energetic disorder and enhanced optical gain property compared with individual perovskite crystal. Secondly, the tunable 2D semiconductor layers cleanly interfaced with perovskites can serve as charge carrier injection layers, which offers potential ready-to-use material options for developing future perovskite-based electronically driven laser. Thirdly, the van der Waals epitaxial halide perovskite/2D semiconductor heterostructures are able to monolithically integrate on both CMOS-compatible substrate (SiO₂/Si) and photonic-compatible platforms (Si and LiNbO₃). This general method offers a diverse and versatile material platform for designing on-chip reconfigurable photonic-electronic integrated devices. Therefore, these advantages and promises motivate us to explore the photonic lasing application of epitaxial heterostructures.

Comment#3-2: The claim of “Threshold is much lower” may not be accurate when comparing 2.21 with 6.36 uJ/cm². The authors may need to reconsider their claims throughout the manuscript, such as “highly reproducible...”, “largely reduced...”, “excellent” and so on.

Authors response: We sincerely thank the reviewer for the very careful reviews and helpful suggestions. Based on the reviewer’s suggestion, we have corrected the corresponding expressions in revised manuscript, where we describe the related contents as quantitatively as possible for maximizing accuracy.

Comment#3-3: Light is a wave with both amplitude and phase. Laser light is coherent. To claim a sharp peak to be lasing, only showing data on amplitude changing with pump power is insufficient. Could the authors provide lasing data on phase, such as the temporal coherence (or spatial coherence), or quantum photon statistics data?

Authors response: We sincerely thank the reviewer for the constructive comments and suggestion. We agree that the reviewer’s suggestion will enhance the robustness of the study finding. In response to the constructive feedback, we comprehensively and quantitatively demonstrated the coherence properties by conducting related experiments and rigorously analyzing the data. In the revised manuscript, we have added the corresponding information in the section of “Reduced laser threshold of

CsPbI₂Br/WSe₂ heterostructure”, Page 16 and 24 in the main text and Figure S16, Page 23-25 in the Supplementary Information.

Figure S16, Page 23-25 in the Supplementary Information

Optical coherence is the predictability of laser wave properties (amplitude and phase) from the process of stimulated emission, due to the generation of photons with “copied” wave traits (phase, wavelength, polarization, and propagating direction) from its spontaneously emitted “seed” photon. Laser light has two kinds of optical coherence, temporal coherence and spatial coherence. (Nature Photonics 3, 546–549 (2009))

Temporal coherence corresponds to the preservation of the phase relationship with time. The necessary conditions for temporal coherence are that all photons should be emitted with same phase and should have the same wavelength. Therefore, spectral narrowing is a basic indicator of the temporal coherence of lasing emission and is usually used as a measure of the degree of coherence (Nature Photonics 14, 375–382 (2020); Advanced Function Materials 26, 6238–6245 (2016)). Figure S16 shows the comparison of the photoluminescence (PL, Figure R5 (a)), amplified spontaneous emission (ASE, Figure S16 (b)) and single-mode lasing (Figure S16 (c)) of CsPbI₂Br/WSe₂ heterostructure. The PL displays a broad and featureless spectrum with full width at half maximum (FWHM) of 32.5 nm, ASE shows a relative narrower spectrum with FWHM of 8.3 nm, while the lasing peak exhibit a very narrow single spectrum with FWHM of 0.7 nm. The monochromatic linewidth below 1.0 nm is well consistent with the emission characteristic of halide perovskite laser, which also suggests that the output light is temporally coherent (Halide perovskite lasers. 1-19 (Springer Nature Singapore, 2022)). The observed linewidth narrowing is also in agreement with the Schawlow–Townes equations, which theoretically predict a sharp decrease of linewidth at the transition from incoherent to coherent emission (Nature Photonics 14, 375–382 (2020)). To further quantitatively assess the degree of the temporal coherence, we estimate the corresponding coherent time (T_c) and length (L_c) as below (Principles of lasers. 475-504 (Springer US, 2010).; Springer Handbook of Lasers and Optics. 583-936 (Springer New York, 2007)).

$$L_c = CT_c$$
$$L_c = \lambda^2 / \Delta\lambda$$

Where the C is the speed of light, λ is the wavelength of lasing peak and $\Delta\lambda$ is the linewidth of the lasing spectrum. The estimated coherent length is as long as 622 μm , which is more than an order of magnitude and six times longer than the state-of-art reports on halide perovskite lasers (20 μm , (Advanced Materials 35, 2302170 (2023)) and 115.6 μm , (Advanced Materials 35, 2306102 (2023,)), respectively. The corresponding coherent time is 0.21 ps. The high temporal coherence performance of epitaxial heterostructure micro-laser can be attributed to self-organized high-quality whispering-gallery-mode (WGM) semiconductor resonator, minimized energetic disorder landscape and high optical gain property.

Spatial coherence measures the correlation of wave phases extending along a

single wavefront, which depends upon the transverse mode discrimination property of the laser resonator. This feature is typically characterized by a well-defined beam with well-defined phase across it and overall beam directionality (Nature Photonics 3, 546–549 (2009); Nature 389, 362–364 (1997)). The experimental observation in Figure 5d in manuscript clearly shows four corner bright lasing output beams when the pump density exceeds the threshold, clearly suggesting the existence of spatial coherence. To estimate the degree of the spatial coherence, the beam directionality emitted by our heterostructures laser (Figure S16 (d)) is assessed by the linear polarization of the lasing emission, as shown in Figure S16 (e) and (f). The degree of polarization (DOP = $(I_{max} - I_{min}) / (I_{max} + I_{min})$, where I_{max} and I_{min} are the maximum lasing intensity and the minimum lasing intensity, respectively), of the single-mode lasing was measured up to 69 %, significantly higher than that of the spontaneous emission peak (~4%). Coupled with the lasing spectra with only one mode, the single lasing mode could be assigned to the fundamental transverse Gaussian beam mode (TEM₀₀) (Advanced Materials 35, 2306102 (2023)). The robust polarization selectivity in the WGM cavity suggests good spatial coherence. The reasons could be attributed to the following reasons. (1) In self-organized perovskite laser, short- and long-range electron–hole exchange interactions result in a splitting of the band-edge excitonic states into an optically inactive singlet state and three optically active triplet states. The competition between three-fold degenerate bright-triplet and a dark singlet transition leads to the linear-polarised outputs (Nature Photonics (2024). <https://doi.org/10.1038/s41566-024-01398-y>). (2) The optical birefringence from Vernier-effect coupling forces single-mode outputs with strong linear polarization properties (Advanced Materials 35, 2302170 (2023)).

The sharp linewidth narrowing, monochromatic lasing output, long-range coherence, high-bright lasing output beam and large output polarization unambiguously demonstrate coherent lasing emission.

Page 16 in the main text

The coherence property is comprehensively assessed by mode narrowing and output polarization, which are inherently associated to temporally and spatially coherent lasing emission. The detailed information is presented in Figure S16. The spectra linewidth is sharply narrowed down to 0.7 nm (<1.0 nm) with above-threshold excitation, which is well consistent with the emission characteristic of halide perovskite laser and with the Schawlow–Townes equations. This clearly signifies the emission transition from incoherent to coherent. The estimated coherent length is as long as 622 μm , which is higher than those reported recently for perovskite-based lasers. The corresponding coherent time is 0.21 ps, strongly suggesting the coherent emission. Furthermore, the degree of polarization of the lasing mode was measured up to 69%, indicating robust polarization selectivity and good spatial coherence. This may originate from the competition between three-fold degenerate bright-triplet and a dark singlet transitions and optical birefringence from Vernier-effect coupling in self-organized perovskite laser. The obvious threshold behavior, sharp linewidth narrowing,

monochromatic lasing output, long-range coherence, high-bright lasing output beam and large degree of emission polarization unambiguously demonstrate the onset of lasing action.

Page 24 in the main text

The polarization ratio was obtained from the lasing spectra recorded at different rotation angles of a polarizer placed in front of the spectrometer.

Figure R5 (Figure S16 in the Supplementary Information): The optical coherence property of lasing action in CsPbI₂Br/WSe₂ heterostructure laser. A comparison of the photoluminescence (PL), amplified spontaneous emission (ASE) and single mode lasing of CsPbI₂Br/WSe₂ heterostructure in (a), (b) and (c), respectively. (d) The optical image of CsPbI₂Br/WSe₂ heterostructure for polarization angle-dependent lasing action.

Scale bar: 15 μm . (e) Intensity polar plot of CsPbI₂Br/WSe₂ lasing output through a rotational analyzer. The experimental data for single mode lasing and spontaneous emission are marked by red dots and black dots, respectively. The solid lines are fits to Malus's law. (f) Lasing spectra of a single CsPbI₂Br/WSe₂ heterostructure pumped above lasing threshold with orthogonal detection polarizations. These detected polarization characteristics demonstrated a linear polarization with degree of polarization (DOP) of 69%, showing a strong output polarization and thus long-range spatial coherence.

REVIEWERS' COMMENTS

Reviewer #1 (Remarks to the Author):

All the requested revisions been adequately addressed and this work is recommended for publication.

Reviewer #2 (Remarks to the Author):

The authors have fully addressed my concerned issues. Therefore, I recommend that the manuscript be accepted for publication.

Reviewer #3 (Remarks to the Author):

The authors have addressed most of the referee's comments, and their efforts are appreciated. For the comment #3-3, two typical literatures may be useful for further investigation into the coherence of a narrow photoluminescence peak, such as Nature 576, 80–84 (2019), and Science Advances. 2019 26; 5(4):eaav4506. Exploration of more literatures on this topic may help the authors in providing sufficient data or discussions on lasing emission.

Manuscript ID: NCOMMS-24-08984A
Point-by-Point Response to Review Comments

Reviewer #1 (Remarks to the Author):

All the requested revisions been adequately addressed and this work is recommended for publication.

Response: We thank the reviewer for acknowledging the acceptance of our work.

Reviewer #2 (Remarks to the Author):

The authors have fully addressed my concerned issues. Therefore, I recommend that the manuscript be accepted for publication.

Response: We thank the reviewer for acknowledging the acceptance of our work.

Reviewer3#-3:

The authors have addressed most of the referee's comments, and their efforts are appreciated. For the comment #3-3, two typical literatures may be useful for further investigation into the coherence of a narrow photoluminescence peak, such as Nature 576, 80–84 (2019), and Science Advances. 2019 26; 5(4):eaav4506. Exploration of more literatures on this topic may help the authors in providing sufficient data or discussions on lasing emission.

Response: We thank the reviewer for the valuable comment. We really appreciated the recommended literatures and cited them within corresponding discussion in the main text (line 27 and line 30 of Page 12) as reference 63 and 65 and Supplementary Information. In response to the reviewer's suggestions, we then explored more related literatures on this topic for providing an in-depth discussion, which is added in Supplementary Figure 16, Page 24 in Supplementary Information.

The good coherence deduced from noticeable linewidth narrowing also rule out the possibility of localized excitons as a source of lasing (Nature 576, 80–84 (2019)). If possible, the higher disorder states in individual perovskite laser should render a lower threshold than that of heterostructure laser, which is inconsistent with our experimental observations. Furthermore, the main factors governed the coherence of laser and improvement methods are discussed accordingly based on the theory of the linewidth of semiconductor lasers (Science Advances. 2019 26; 5(4):eaav4506; Adv. Optical Mater. 6, 1800272 (2018)). In general, the width of the laser linewidth mainly originates from the fluctuations in the phase of the optical field. On the one hand, these fluctuations arise from spontaneous emission events, which discontinuously alter the phase and intensity of the lasing field. On the other hand, the change in refractive index due to the change of carrier density can results in an additional phase shift of the laser field and in additional line broadening. The unified theory for qualifying the linewidth (Δf) is expressed as follow (IEEE J. Quantum Electronic. 18, 259 – 264, (1982)):

$$\Delta f = \frac{u_g^2 h \nu g n_{sp} \alpha_m (1 + \alpha^2)}{8\pi P_0}$$

where u_g is the group velocity, $h\nu$ is the energy of the output laser, g is the modal gain, n_{sp} is referred to as the spontaneous emission factor, α_m is the facet loss factor, $1 + \alpha^2$ is enhancement factor, P_0 is the output power. Based on the equation, three strategies can be inspired to compress the linewidth and thus enhance the coherence (Nat. Rev. Phys. 1, 156–168 (2019)). Firstly, reducing cavity loss caused by facet and waveguide losses, which can be achieved by embedding the semiconductor nanostructures into other ultra-low loss optical cavities such as silica whispering-gallery-mode cavities or distributed Bragg reflector cavities. Secondly, increasing the photon lifetime within the cavity can suppresses intensity fluctuations of the emission, enhancing the second-order coherence. This can be realized by tailoring the spatial structures of lasing modes and their nonlinear interactions within the gain material. Finally, enhancing the output power might be viable path to regulate the coherence by integrating an external cavity for underpinning the feedback modal.